# Species -shared and -unique gyral peaks on human and macaque brains

**Songyao Zhang[1], Tuo Zhang[1]\*, Guannan Cao[1], Jingchao Zhou[2], Zhibin He[1], Xiao Li[3], Yudan Ren[3], Tao Liu[4], Xi Jiang[2], Lei Guo[1], Junwei Han[1], Tianming Liu[5]**

[1]School of Automation, Northwestern Polytechnical University, Xi'an, China; [2]School of Life Science and Technology, MOE Key Lab for Neuroinformation, University of Electronic Science and Technology of China, Chengdu, China; [3]School of Information Technology, Northwest University, Xi'an, China; [4]College of Science, North China University of Science and Technology, Tangshan, China; [5]Cortical Architecture Imaging and Discovery Lab, Department of Computer Science and Bioimaging Research Center, University of Georgia, Athens, United States

**\*For correspondence:**
tuozhang@nwpu.edu.cn

**Competing interest:** The authors declare that no competing interests exist.

**Abstract** Cortical folding is an important feature of primate brains that plays a crucial role in various cognitive and behavioral processes. Extensive research has revealed both similarities and differences in folding morphology and brain function among primates including macaque and human. The folding morphology is the basis of brain function, making cross-species studies on folding morphology important for understanding brain function and species evolution. However, prior studies on cross-species folding morphology mainly focused on partial regions of the cortex instead of the entire brain. Previously, our research defined a whole-brain landmark based on folding morphology: the gyral peak. It was found to exist stably across individuals and ages in both human and macaque brains. Shared and unique gyral peaks in human and macaque are identified in this study, and their similarities and differences in spatial distribution, anatomical morphology, and functional connectivity were also dicussed.

## eLife assessment

This **important** paper compares cross-species cortical folding patterns in human and non-human primates, showing that most gyral peaks shared across species are in lower-order cortical regions. The supporting evidence is **solid** and multi-faceted, encompassing anatomy, connectivity and gene expression. This paper will be of interest to a broad readership within the neuroscience community, especially for those interested in cross-species correspondences in brain organisation.

## Introduction

Humans and macaques share a common ancestor, but they have diverged evolutionarily approximately 25 million years ago (*Hill et al., 2010*). As a result of genetic changes, environmental factors, and selective pressures (*Lecouvet et al., 1997*), they have developed distinct brain structures and functions. Cortical folds are important features of primate brains. The primary driver of cortical folding is the differential growth between cortical and subcortical layers. During the gyrification process in the cortex, areas with high-density stiff axonal fiber bundles towards gyri. The folding patterns in the brain, formed through a series of complex processes, are found to play a crucial role in various cognitive and behavioral processes, including perception, action, and cognition (*Fornito et al., 2004*; *Cachia et al., 2018*; *Yang et al., 2019*; *Whittle et al., 2009*).

Studies have revealed differences and similarities in fold morphology and brain function between humans and macaques (*Semendeferi et al., 2002*). Furthermore, there is a intricate relationship between the similarities and differences in cortical folding morphology and the similarities and differences in brain function. For example, humans possess a larger prefrontal cortex compared to macaques, which gives them executive functions such as planning, decision-making, and working memory (*Semendeferi et al., 2002*). The higher cognitive and affective functions observed in human compared to macaque are also associated with the larger proportion of their association cortex in the cortical surface (*Glasser et al., 2013*; *Rilling, 2014*; *Rolls and Grabenhorst, 2008*). The variations in cortical folding morphology, as well as the differences in brain function, may reflect the adaptation of species to diverse cognitive, social, and ecological demands. Despite variations in morphological and functional characteristics of the cortical folds among different species, there are also many commonalities, indicating relative conservation in the evolutionary process (*Van Essen et al., 2019*). For example, gyrencephalic primates which share many primary sulci, such as the lateral, superior temporal, and (except for the marmoset) central sulci, exhibit similarities in both morphology and brain function (*White et al., 1997*; *Ferrier, 1873*; *Friedrich et al., 2021*). Additionally, by comparing the brain activity of chimpanzees during tasks with nonsocial tasks and at rest, it was found that the cortical midline areas of chimpanzees deactivate during these tasks. This suggests that the DMN of chimpanzees is anatomically and functionally similar to that of humans (*Barks et al., 2015*). Some studies have found that in species including humans and monkeys, strongly interconnected regions are consistently separated by outward folds, whereas weakly connected regions are consistently separated by inward folds. This folding pattern is associated with brain connectivity, suggests a certain similarity in the mechanisms underlying cortical folding in humans and monkeys (*Essen, 1997*; *Sereno et al., 1995*; *Sousa et al., 1991*). In summary, the folding patterns and functional profiles of cortical regions demonstrate both similarities and differences across different species. These similarities may reflect evolutionary conserved functions, while the differences may indicate species-specific features (*de Lange et al., 2019*; *Buckner and Krienen, 2013*; *Patel et al., 2015*).

Most of the current cross-species studies are based on one or several anatomical landmarks (*Eichert et al., 2019*; *Goulas et al., 2014*; *Van Essen and Dierker, 2007*; *Van Essen et al., 2018*), which cannot really solve the cross-species analysis needs of the whole brain. The definition of a whole-brain anatomical landmark across species is a complex task due to differences in brain size, cortical folding patterns, and the relative size and location of different brain regions. As a landmark defined based on cortical fold pattern, the gyral peak, as the maximum in height on the cortex, has been discovered and studied in both humans and macaques (*Zhang et al., 2022*; *Zhang et al., 2023*). It is defined as the local highest point of the gyri. In the previous work, there are many similarities between the findings of humans and macaques regarding gyral peaks. For example, both species were able to detect consistent gyral peaks among individuals on the cerebral cortex. And it was even consistent across ages in the longitudinal macaque dataset (*Zhang et al., 2022*). In both of these works, there is a discussion of peaks' anatomical feature and inter-individual consistency. In the study of macaques, it has been observed that the peak consistently present across individuals is located on more curved gyri (*Zhang et al., 2022*). Similar conclusions have been drawn in human brain research (*Zhang et al., 2023*). While some findings are not entirely the same between humans and macaques. For example, the higher consistency peaks in humans possessing smaller structural connectivity properties, while the conclusion is opposite in macaques. In addition, there are some conclusions that have been verified on only one species. Based on the aforementioned advantages of gyral peaks, they are highly suitable as anatomical landmarks for cross-species research to infer the developmental and evolutionary aspects of cortical folding and brain functionality.

In this investigation, shared and unique peak clusters across individuals and species were examined. The group-wise peak clusters of human and macaque brains were identified, respectively, and the macaque peak clusters were aligned onto the human brain surface using cross-species registration methods (*Xu et al., 2020*). This allowed us to identify the shared and unique peak clusters between the two species. Then, the inter-individual consistency of shared and unique clusters within each species was compared, and investigated whether there was a relationship between the inter-individual consistency of shared clusters between human and macaque. Additionally, the anatomical features of these shared and unique clusters were examined, and the functional and structural connectivity matrices of the human and macaque brains were calculated. The Brain Connectivity Toolbox (BCT,

*Rubinov and Sporns, 2010*) was utilized to compute the node features of shared and unique clusters. Furthermore, spatial relationships between these clusters and different brain regions of multiple atlases were explored. Finally, human brain RNA-seq data was employed to select important genes from shared and unique peaks in classification tasks. The significance of this study is to provide a medium based on cortical folding patterns for cross-species cortical analysis. Through such a medium, exploration into the derivation and specialization of human and macaque brain can be facilitated, aiding in understanding the rules of how the brain is constructed during development and evolution (*Krubitzer, 2007*).

## Results

### Locations of shared and unique peak clusters

To obtain shared and unique gyral peaks between species, peak clusters were first extracted for each species. The definition of peaks and the method for extracting peak clusters within each species are described in the Materials and methods section. Subsequently, a cross-species registration method (*Xu et al., 2020*) was applied to align the macaque peak clusters onto the human brain surface. *Figure 1a* top and middle panels display the locations of all peak clusters found in both human and macaque brains (Human: LH-96, RH-96; Macaque: LH-42, RH-43). Then the cross-species registration method (*Xu et al., 2020*) was utilized to register the peak clusters of the macaque brain onto the human brain surface (*Figure 1a* bottom panel uses the same color-coding as the macaque surface to represent the same cluster). Next, based on the definition of shared peak clusters (see Materials and methods), shared and unique gyral peaks between the two species were identified. *Figure 1b* shows the locations of shared peak clusters between the two species, with 25 shared peaks in the left hemisphere while 26 in the right (the locations of all human shared peaks are reported in *Table 1*). For the purpose of comparison, the shared gyral peak clusters of two species were displayed on the surface of the human brain template (Conte69, *Van Essen et al., 2012b*) with the same color coding for corresponding peak clusters on the two species. The results of shared peak clusters on the macaque surface template are placed in Supplementary Information. *Figure 1c* shows the locations of unique peak clusters found in each species, with 141 (LH-71, RH-70) unique peak clusters found in the human brain and 34 (LH-17, RH-17) found in the macaque brain. The unique peaks found in the human brain were mapped onto the Conte69, while those found in the macaque brain were mapped onto the Yerkes19 (*Van Essen et al., 2012a*) template surface. It is worth noting that for each species, the union of the clusters in *Figure 1b* and *Figure 1c* is the same as the clusters in *Figure 1a* (including color).

To investigate the regions where shared and unique peaks are located, the Cole-Anticevic Brain-wide Network Partition (CA network, *Ji et al., 2019*) was utilized, which includes in total 12 functional networks (*Figure 2a* right panel) based on the MMP (multimodal parcellation, *Glasser et al., 2016*). The human Cole-Anticevic network was projected onto the macaque surface using the method described by *Xu et al., 2020*. This allowed for a qualitative comparison of the differences in the distribution of cluster centers between human and macaque. It is important to emphasize that while the shared peak clusters were obtained through cross-species registration, and the human brain network (Cole-Anticevic) was also transferred to the macaque surface using cross-species registration, it is still meaningful to compare the distribution of shared peak centers between humans and macaques. This is because the intersection of clusters (one of the definition of shared peak clusters) does not necessarily imply that the centers of peak clusters are located in the same brain region.

The number of shared and unique peaks distributed in different brain networks was counted (*Figure 2a*). In human brain, most shared peak cluster centers are distributed in the networks such as somatomotor (SMN), visual 1 (V1), and visual 2 (V2), while most human unique peak cluster centers are located in the networks such as default-mode network (DMN), cingulo-opercular (CON), and frontoparietal (FPN). In the macaque brain, shared peak cluster centers most distributed in the V2, DMN, and CON, while unique peak cluster centers most distributed in the higher-order networks such as DMN, language (Lan), and dorsal attention (DAN). In order to eliminate the influence of different brain area sizes on the count of peak cluster, normalization by the regional surface area was conducted (*Appendix 3—figure 2*).

In general, to clarify the distribution of shared and unique peaks in the high-order and low-order networks, the 12 brain networks in Cole-Anticevic atlas were divided into the low-order networks

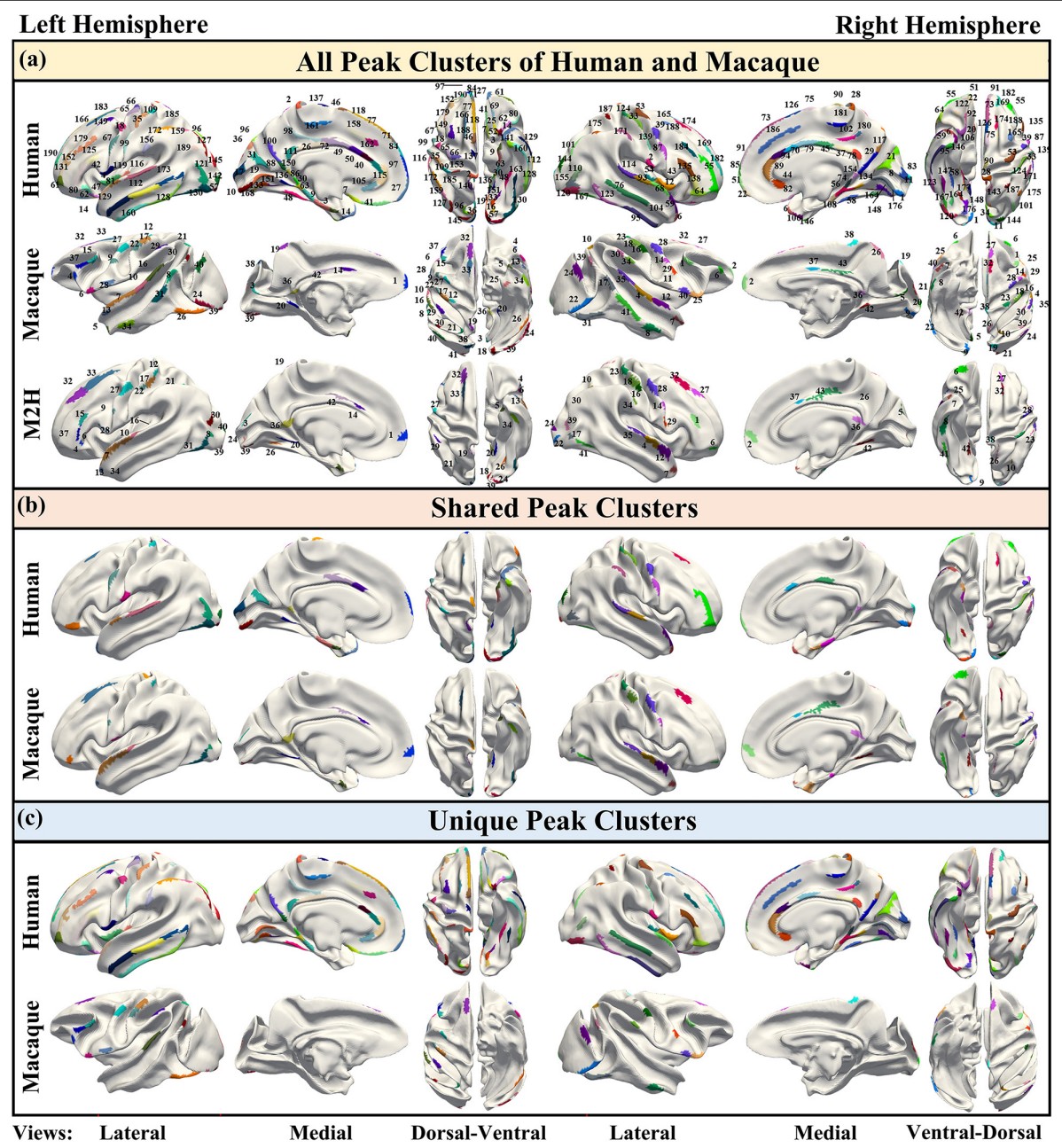

**Figure 1.** The spatial distribution of all gyral peaks of human and macaque, as well as their shared and unique gyral peaks. (a) Top: 192 gyral peak clusters of human on human brain template (Conte69, *Van Essen et al., 2012b*). Middle: 85 gyral peak clusters of macaque on macaque brain template (Yerkes19, *Van Essen et al., 2012a*). Bottom: The results of mapping macaque gyral peak clusters on the human brain template by the cross-species registration (*Xu et al., 2020*). The same color of middle and bottom surface indicates the corresponding peak clusters. (b) Peak clusters shared by human and macaque (LH-25, RH-26). On the same hemisphere of the brain, the corresponding-colored regions on both human and macaque represent the corresponding shared peak clusters. In addition, the color of the left and right hemisphere clusters are not related. (c) Unique peak clusters of two species map on the surface of their respective surface template.

(visual 1 (V1), visual 2 (V2), auditory (Aud), somatomotor (SMN), posterior multimodal (PMN), ventral multimodal (VMN), and orbito-affective networks (OAN)) and higher-order networks (include cingulo-opercular (CON), dorsal attention (DAN), language (Lan), frontoparietal (FPN), default mode network (DMN)) based on previous research (*Golesorkhi et al., 2022*; *Ito et al., 2020*). On this lower/higher order division, the number of shared and unique peaks in both species was reported in *Table 1*. *Figure 2a* and *Table 1* collectively indicate a conclusion: whether in humans or macaques, shared

**Table 1.** The number of shared and unique peaks in lower- and higher-order brain networks of the two species.

Lower-order networks include visual 1 (V1), visual 2 (V2), auditory (Aud), somatomotor (SMN), posterior multimodal (PMN), ventral multimodal (VMN), and orbito-affective networks (OAN), higher order networks include cingulo-opercular (CON), dorsal attention (DAN), language (Lan), frontoparietal (FPN), default-mode network (DMN).

| Lower/Higher cortex | Human | Macaque |
|---|---|---|
| Shared peak | 33/18 | 29/22 |
| Unique peak | 37/104 | 14/20 |

peaks are more distributed in lower order networks, while unique peaks are more in higher order networks. This observation is particularly pronounced in humans.

While it is known where shared and unique peaks are distributed across different brain networks, the dominance of each type of peak within each networks remains unrevealed. *Figure 2b* reports the ratio between peak count and unique peak count for each network, such that the networks where the most shared or unique peaks are found can be easily highlighted. To mitigate potential imbalances in proportions caused by differences in the absolute numbers of each category (shared or unique) of peak, the proportions of peaks within their respective categories were utilized in the calculations. The pink and green color bins represent ratios of shared and unique peaks, respectively. The dark blue dashed line represents the 50% reference line. In general, from left to right in the figure, the ratio of shared peaks decreases gradually while the ratio of unique peaks increases, suggesting that shared peaks are more (>0.5, above the dashed line) on lower order networks (orange font), while unique peaks are generally more on higher order networks (blue font). In specific, in human brains, the brain networks with a higher abundance of shared peaks are Aud, VMN, V1, SMN, and V2; whereas in macaques, they are CON, VMN, V1, V2, FPN, and SMN. Again, in the human brains, the disparity between shared and unique peaks tends to be more significant (further away from the reference line), for both lower order and higher order networks, respectively. In contrast, in the macaque brains, the disparity between shared and unique peaks is less significant (closer to the reference line). The ratio of shared and unique peaks is around 0.5 for 6 out of all 10 networks (including both lower and higher order ones).

## Consistency of unique/shared peak clusters

In the previous researches, the inter-individual consistency of peaks is a measure to assess whether peaks exist consistently in different individuals (*Zhang et al., 2022*; *Zhang et al., 2023*). To explore the inter-individual consistency of shared and unique peak clusters in macaques and humans, the mean count covered by these clusters was calculated and normalized by the number of individuals, as presented in *Figure 3a*. In both human and macaque, the consistency of shared peak clusters is significantly greater than that of unique peak clusters (p<0.001, t-value=4.74 for human and 2.67 for macaque). Additionally, the overall consistency of peaks in the macaque brain is much higher than that in the human brain, indicating that the peaks in the different macaque brain are more concentrated in spatial distribution. Furthermore, linear regression analysis was performed on the average counts of all corresponding shared peak clusters of human and macaque. The horizontal and vertical axes of the *Figure 3b* represent the average count of shared peaks in the macaque and human brains, respectively. The Pearson correlation coefficient (PCC) of the inter-species consistency of the left and right brain is 0.20 and 0.26 (p>0.05 for all), respectively. The result of linear regression shows that there is a positive correlation in the inter-individual consistency of shared peaks between macaque and human brains, but it is not statistically significant (with $R^2$ for the left and right brain are 0.07 and 0.01, respectively).

## Anatomical features of shared and unique peaks

The mean of the anatomical features of shared and unique peaks across all individuals of both species was calculated. The shared and unique peaks in each individual were obtained by calculating the intersection between the group-wise shared and unique clusters and the gyral peaks in each individual. It

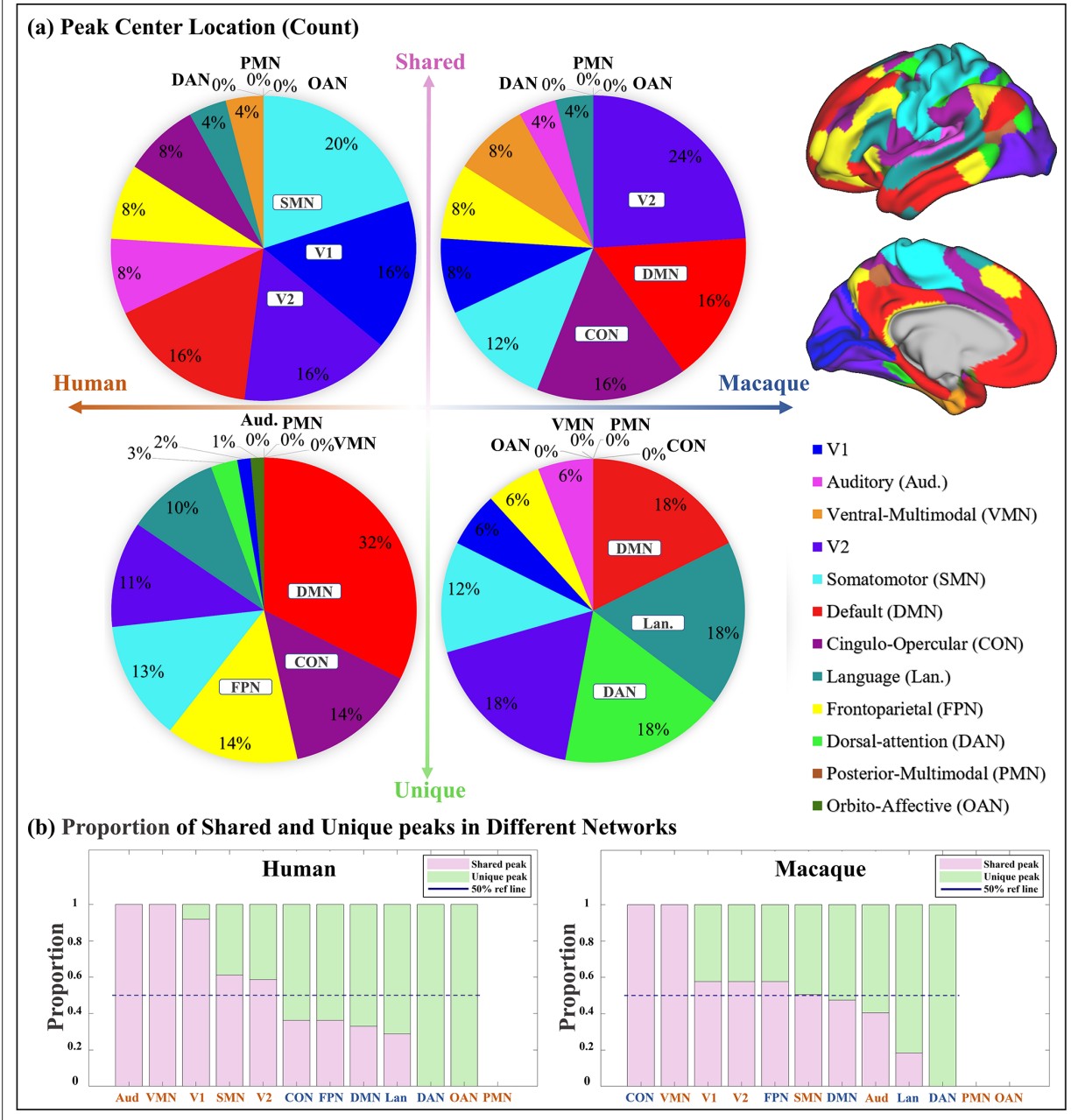

**Figure 2.** Spatial distribution characteristics of shared and unique gyral peaks. (**a**) Pie chart shows the count of shared and unique peaks across different brain networks for both human and macaque. Right panel shows the Cole-Anticevic (CA) networks (*Ji et al., 2019*) on human surface as a reference. (**b**) The ratio of shared and unique peaks in each brain network in the Cole-Anticevic (CA) atlas. The pink and green color bins represent ratios of shared and unique peaks, respectively. The dark blue dashed line represents the 50% reference line. For each brain region, the sum of the ratios of shared and unique peaks is equal to 1.

was found that, in both human or macaque, the sulcs and local surface area of shared peaks are larger than those of the unique peaks, but the curvatures are smaller. Due to issues with MRI data quality and technical limitations, only the white matter surface of the macaque brain was reconstructed, and the gray matter surface was not available. Therefore, it was not possible to calculate cortical thickness for the macaque dataset. Additionally, due to the unavailability of T2 data in the macaque dataset, the myelin feature was also missing. For the exclusive anatomical features of human, shared gyral peaks are located in cortical regions with thinner cortex but larger myelin in contrast of unique peaks

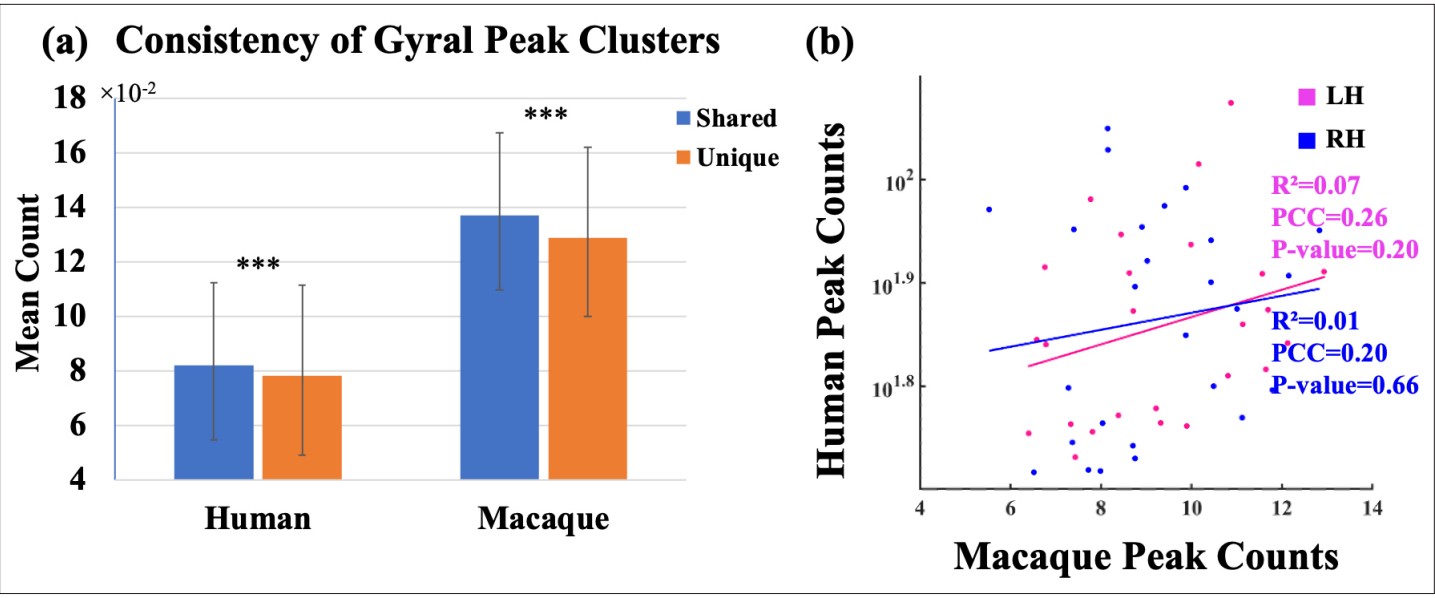

**Figure 3.** Consistency results of shared and unique peaks of two species. (**a**) Mean peak count (± SD) covered by shared and unique peak clusters in two species. ***indicates p<0.001. The t-values for the t-tests in humans and macaques are 4.74 and 2.67, respectively. (**b**) Linear regression results of the consistency of peak clusters shared between macaque and human brains. The pink and blue colors represent the left and right hemispheres, respectively. The results of the linear regression are depicted in the figure. While there was a positive correlation observed in the consistency of gyral peaks between macaque and human, the obtained p-value for the fitted results exceeded the significance threshold of 0.05.

(*Table 2*). The statistical analysis conducted using t-tests revealed that the p-values for shared and unique peaks of all features were less than 0.001, except for the local surface area of the macaque.

## Functional connectivity characteristics of shared and unique peaks

*Table 3* shows the mean (± SD) of node properties of the functional connectivity for all shared and unique peak clusters in human and macaque, including degree, strength, clustering coefficient (CC), betweenness, and efficiency. In general, the results demonstrate that shared peaks exhibit significantly (p<0.001) larger degree, strength, clustering coefficient, betweenness, and efficiency values than unique peaks (except for betweenness and efficiency of macaque) for the functional connectivity characteristics. The mean values of all node properties, as well as the p-values and t-values of the t-test between shared and unique peaks, are all displayed in *Table 3*. In addition, a comparison was made between shared and unique peaks on the structural connectivity matrix of the human brain, and the results are presented in *Appendix 6—table 1* of the Supplementary Information (due to the poor tracking effect of dti fiber tractography in the macaque data, only the structural connection matrix of human brain was calculated).

**Table 2.** The mean (± SD) of anatomical features, as well as the p-values and t-values of the t-test between shared and unique peak clusters.
In the t-test, n for human is 880 and for macaque is 591. The bold font is the one with the larger values of shared and unique peaks.

| | Human | | | | | Macaque | | |
|---|---|---|---|---|---|---|---|---|
| | Sulc | Curv | Myelin | Thickness | Area | Sulc | Curv | Area |
| Shared | 0.93±0.05 | 0.31±0.02 | 1.85±0.10 | 2.71±0.14 | 1.19±0.09 | 0.86±0.03 | 0.55±0.03 | 0.94±0.44 |
| Unique | 0.79±0.05 | 0.32±0.01 | 1.83±0.11 | 2.94±0.12 | 1.09±0.05 | 0.80±0.04 | 0.58±0.03 | 0.91±0.17 |
| p | <0.001 | <0.001 | <0.001 | <0.001 | <0.001 | <0.001 | <0.001 | 0.59 |
| t | 58.43 | −16.26 | 6.51 | −36.67 | 30.43 | 6.07 | −5.32 | 0.54 |

**Table 3.** The mean (± SD) functional connectivity characteristics, as well as the p-values and t-values of the t-test between shared and unique peak clusters of human and macaque.

In the t-test, n for human is 880 and for macaque is 591. The bold font represent the larger values between the shared peak and unique peaks.

|  |  | Degree | Strength | CC | Betweeness | Efficiency |
|---|---|---|---|---|---|---|
| Human FC | Shared | 141.13±30.46 | 52.27±22.84 | 0.20±0.07 | 1.87±0.74(×$10^3$) | 0.25±0.07 |
|  | Unique | 119.88±18.03 | 44.35±15.24 | 0.19±0.05 | 1.46±0.43(×$10^3$) | 0.24±0.06 |
|  | p | <0.001 | <0.001 | <0.001 | <0.001 | <0.001 |
|  | t | 7.78 | 5.24 | 3.94 | 4.42 | 3.37 |
| Macaque FC | Shared | 136.60±21.89 | 43.74±8.85 | 0.18±0.05 | 2.00±0.50(×$10^3$) | 0.25±0.07 |
|  | Unique | 134.69±23.51 | 43.30±8.15 | 0.17±0.05 | 2.18±0.60(×$10^3$) | 0.24±0.07 |
|  | p | <0.01 | <0.001 | <0.01 | <0.001 | >0.05 |
|  | t | 2.98 | 5.01 | 2.64 | –6.52 | 0.53 |

## Spatial relationship between peaks and functional regions

To assess the relative spatial relationship between the two types of peaks and different brain regions, the number of brain regions where each type of peak appeared within a 3-ring neighborhood was calculated. Various types of brain atlases were utilized, including those based on functional, structural, and cytoarchitectural. These atlases are crucial because they contain diverse features of the brain, helping to identify spatial patterns of shared and unique peaks across multiple references. *Table 4* and *Table 5* present the results for 10 human brain atlases and 3 macaque brain atlases, respectively (results of all other human atlases are presented in *Appendix 7—table 1*). False discovery rate (FDR) correction was utilized for multiple comparisons, and the corrected p-values are reported in tables (n=880 for human and n=591 for macaque). The observation that more diverse brain regions around shared peaks than around unique peaks for multiple brain atlases with a median parcellation resolution (7 parcels to 300 parcels), demonstrating the robustness of the conclusion.

## Gene analysis of shared and unique peak clusters based on Lasso

Finally, to study whether there are significant differences in gene expression between the two types of peaks, the surface-based gene expression dataset Allen Human Brain Atlas (AHBA) (*Arnatkeviciute et al., 2019*, *Hawrylycz et al., 2012*) was utilized, employing the widely used lasso method for gene selection. The preprocessed AHBA gene data is in the form of region×gene and the region above referred to the parcellation of a certain atlas, such as Aparc, Schaefer100, Schaefer500, Schaefer1000,

**Table 4.** The mean values (± SD) of brain regions that appeared within a 3-ring neighborhood for shared and unique peaks in 10 common human atlases.

All the shared peaks in the table have a greater number of neighboring brain regions compared to the unique peaks. All p<0.001, false discovery rate (FDR) corrected.

| Atlas Name | Glasser2016 | Schaefer-100 | Schaefer-200 | Schaefer-300 | Vosdewael-100 |
|---|---|---|---|---|---|
| Share Nbr | 2.43±0.15 | 1.89±0.12 | 2.12±0.11 | 2.23±0.11 | 1.57±0.17 |
| Unique Nbr | 2.37±0.09 | 1.74±0.09 | 2.08±0.10 | 2.17±0.10 | 1.46±0.10 |
| p | <0.001 | <0.001 | <0.001 | <0.001 | <0.001 |
| t | 8.32 | 26.66 | 4.50 | 18.08 | 34.09 |
| Atlas Name | Yeo2011(17) | Aparc | Aparc2009 | BA | Cole-Anticevic |
| Share Nbr | 1.76±0.11 | 1.58±0.12 | 1.95±0.13 | 1.58±0.12 | 1.65±0.11 |
| Unique Nbr | 1.73±0.08 | 1.33±0.07 | 1.94±0.09 | 1.29±0.08 | 1.57±0.07 |
| p | <0.001 | <0.001 | <0.001 | <0.001 | <0.001 |
| t | 22.29 | 56.37 | 3.80 | 69.84 | 22.44 |

**Table 5.** The mean values (± SD) of brain regions that appeared within a 3-ring neighborhood for shared and unique peaks in 3 common macaque atlases.

For both Markov91 and Cole-Anticevic atlas, the shared peaks has more variety of functional regions around it than the unique peaks. But for the altas BA05, the conclusion was reversed. The bold font represent the larger values between the shared peak and unique peaks. All p<0.001, false discovery rate (FDR) corrected.

| Atlas Name | Markov91 | Cole-Anticevic | BA05 |
|---|---|---|---|
| Share Nbr | 2.73±0.27 | 1.77±0.17 | 1.61±0.16 |
| Unique Nbr | 2.16±0.15 | 1.58±0.16 | 1.80±0.16 |
| p | <0.001 | <0.001 | <0.001 |
| t | −7.4 | 14.93 | 6.49 |

etc. Finally, the Schaefer-500 atlas was selected for this study because high resolution may result in some areas with no gene data (more details refer to *Arnatkeviciute et al., 2019*), while low resolution may result in multiple categories of clusters being located in the same region. Therefore, Schaefer500 was chosen as the most suitable atlas for the analysis. Before using lasso for feature selection, the determination of the Lambda parameter is necessary to regulate the number of selected features. For parameter selection, 10-fold cross validation was employed. By considering the maximization of accuracy (acc) and minimization of mean squared error (MSE) simultaneously, the lambda value was ultimately determined to be 0.027 *Figure 4* (*Figure 5b*). The accuracy of training set was 0.84, and the MSE was 0.64; The accuracy of test set was 0.75, and the MSE was 1.00. Finally, the Lasso method was utilized to select 28 genes with significant impacts on the classification of shared and unique peaks. Then the Welch's t-tests was performed to compare the expression of the 28 genes in the shared and unique peak clusters. The gene list and the corresponding Welch's t-tests results were shown in *Appendix 8—table 1*. Ultimately, seven genes showed significant differential expression between shared and unique peaks. These genes were PECAM1, TLR1, SNAP29, DHRS4, BHMT2,

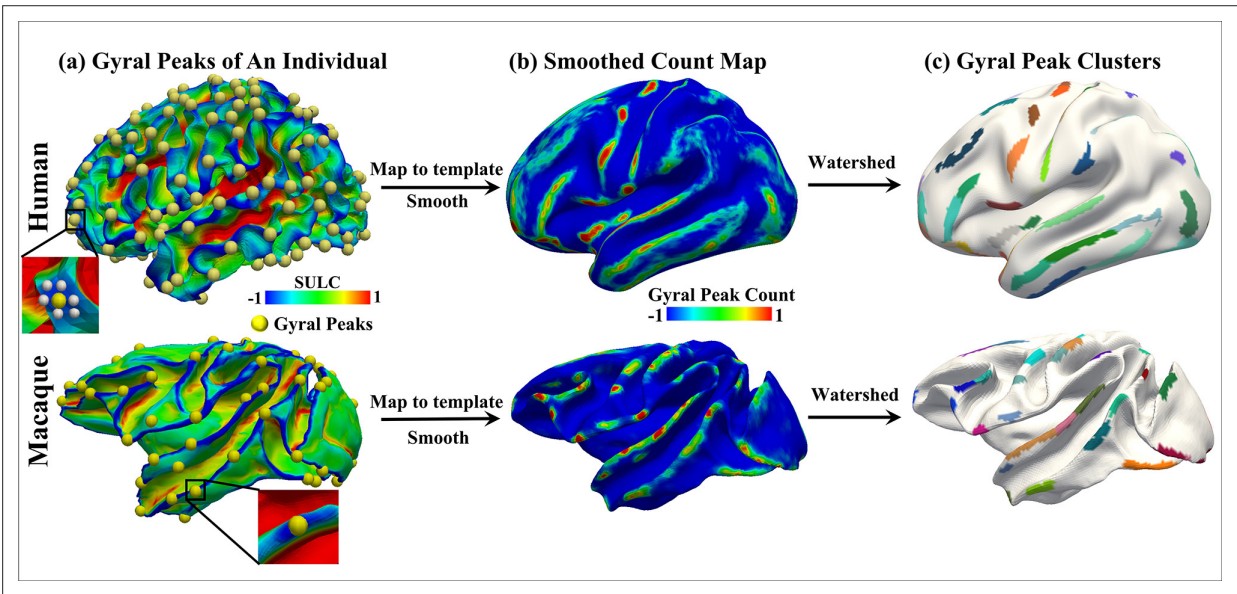

**Figure 4.** Peak cluster extraction pipeline. The two rows represent the human brain and the macaque brain, respectively. (**a**) Shows the locations of all extracted peaks in an individual. (**b**) Due to resampling of the human and macaque surface, there is a vertex-to-vertex correspondence between individuals. Therefore, all individual peaks were placed on the template brain surface and undergo isotropic smoothing, resulting in the count map shown in (**b**), where the highlighted regions indicate a higher frequency of peak occurrences across individuals. (**c**) shows the results of clustering the count map using watershed algorithm, resulting in peak clusters for both species. A total of 192 peak clusters were detected in the human brain, while 85 peak clusters were detected in the macaque brain.

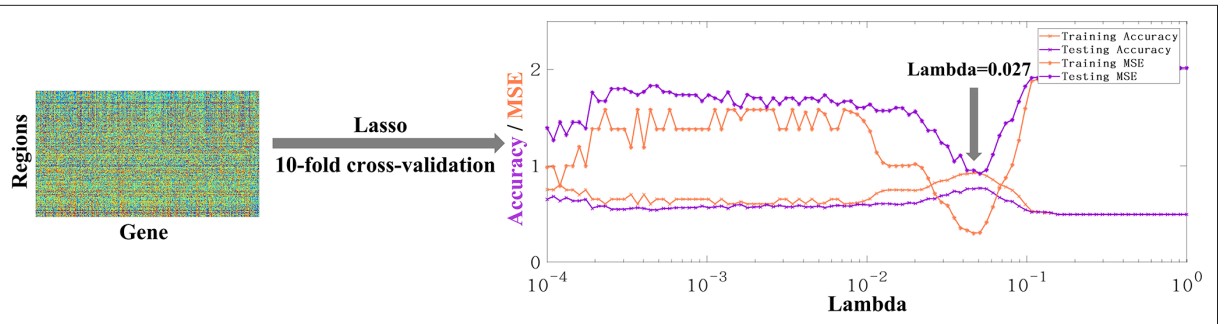

**Figure 5.** The original form of AHBA data is region × gene. The accuracy and MSE line charts of the training set and testing set corresponding to lambda from $10^{-4}$ to 1. Purple and orange respectively represent the accuracy and mse obtained by 10-fold cross verification. The final lambda determined is 0.027, which can ensure the maximum accuracy and minimum MSE at the same time.

PLBD1, KCNH5. Brief descriptions of their functions are listed in *Table 6*. All gene functions descriptions were derived from the NCBI website (https://www.ncbi.nlm.nih.gov/).

## Discussion

In this study, 192 gyral peaks were detected in the human brain, and 85 gyral peaks were detected in the macaque brain. Additionally, 51 pairs of shared peaks (25 in the left and 26 in the right hemisphere)

**Table 6.** Seven genes were selected using LASSO that showed significant differential expression in shared and unique peaks.

| Gene symbol | Gene function |
| --- | --- |
| PECAM1 | The protein encoded by this gene is found on the surface of platelets, monocytes, neutrophils, and some types of T-cells, and makes up a large portion of endothelial cell intercellular junctions. The encoded protein is a member of the immunoglobulin superfamily and is likely involved in leukocyte migration, angiogenesis, and integrin activation. [provided by RefSeq, May 2010] |
| TLR1 | The protein encoded by this gene is a member of the Toll-like receptor (TLR) family which plays a fundamental role in pathogen recognition and activation of innate immunity. They recognize pathogen-associated molecular patterns (PAMPs) that are expressed on infectious agents, and mediate the production of cytokines necessary for the development of effective immunity. [provided by RefSeq, Jul 2008] |
| SNAP29 | This gene, belonging to the SNAP25 gene family, encodes a protein involved in various membrane trafficking processes. Other members of this gene family, such as SNAP23 and SNAP25, encode proteins that bind to a syntaxin protein and facilitate the docking and fusion of synaptic vesicle membranes with the plasma membrane. [provided by RefSeq, Jul 2008] |
| DHRS4 | Exhibits protein binding and oxidoreductase activities, involved in cellular metabolic processes including ketone metabolism, regulation of reactive oxygen species, and steroid metabolism. Found in the nucleus and peroxisomal membrane. [provided by Alliance of Genome Resources, Apr 2022] |
| BHMT2 | Homocysteine, a sulfur-containing amino acid, is crucial for methylation reactions. The protein encoded by this gene is one of two methyltransferases that facilitate the transfer of a methyl group from betaine to homocysteine. Irregularities in homocysteine metabolism have been linked to conditions ranging from vascular disease to neural tube birth defects. This gene has alternatively spliced transcript variants encoding different isoforms.[provided by RefSeq, May 2010] |
| PLBD1 | Predicted to enable phospholipase activity. Predicted to be involved in phospholipid catabolic process. Located in extracellular space. [provided by Alliance of Genome Resources, Apr 2022] |
| KCNH5 | This gene encodes a member of voltage-gated potassium channels. Members of this family have diverse functions, including regulating neurotransmitter and hormone release, cardiac function, and cell volume. This protein is an outward-rectifying, noninactivating channel. Alternative splicing results in multiple transcript variants. [provided by RefSeq, Jul 2013] |

were identified using cross-species registration, as previously reported by *Xu et al., 2020*. The following findings were observed:

1. Spatial distribution: Whether in humans or macaques, shared peaks are more predominantly distributed in lower-order networks, while unique peaks are more prevalent in higher-order networks. This conclusion is particularly pronounced in humans.
2. Consistency: The inter-individual consistency of shared peaks within each species was greater than that of unique peaks. The consistency of shared peaks in the human and macaque brains exhibits a positive correlation (not-significant though).
3. Anatomy: In both human and macaque, it can be found that the sulcs and local surface area of shared peaks are larger but the curvatures are smaller compared to unique peaks in each species. Furthermore, for the exclusive anatomical features of human, shared gyral peaks are located in cortical regions with thinner cortex but larger myelin in contrast of unique peaks.
4. Brain connectivity: Shared peaks in the structural (human only) and functional (human and macaque) graph metrics exhibited higher values for degree, strength, clustering coefficient, betweeness and efficiency compared to unique peaks in both species (except for betweeness and efficiency of the macaque).
5. Relationship with brain regions: Across multiple brain atlases in both species, shared peaks, compared to unique peaks, were found in clusters encompassed by a more diverse array of brain regions.
6. Gene analysis: Employing the Lasso method for feature selection on all genes, it was discovered that some genes related to brain function played an important role in the classification of shared and unique peaks.

It was observed that, whether in humans or macaques, shared peaks are more distributed in lower-order networks, while unique peaks are more in higher order networks (*Figure 2* and *Table 1*). This observation is particularly pronounced in humans. This finding is in line with previous conclusions that cortical regions associated with motor and sensory functions are relatively conserved across species (*Hopkins et al., 2014*; *Krubitzer, 2007*; *Xu et al., 2020*; *Teissier and Pierani, 2021*), while the unique peaks on human appear in brain regions that are specific to human, such as language areas. One possible explanation is the disproportionate expansion of multiple, distributed regions of association cortex relative to sensory regions during species evolution (*Krubitzer, 2007*; *Buckner and Krienen, 2013*). The expansion of these regions, which untethered them from the constraints of sensory hierarchies and established species-specific functional associations, is the foundation of the 'tethering hypothesis' (*Buckner et al., 2013*). Evolutionary psychology and neuroscience indicate that this differential regional allometric growth arises from developmental constraints and represents an adaptive adjustment by the brain to optimize its functional organization (*Montgomery, 2013*; *Montgomery et al., 2016*; *Willemet, 2015*). Based on these studies, interspecies conservation of sensorimotor regions and uniqueness of higher order brain regions are easily understood, and our work provides additional supports to this viewpoint by examining cortical folding.

In each pie chart of *Figure 2a*, the top three ranked brain networks in both species were specifically highlighted. Although the pie chart also generally supports the above results, two brain networks deserve further discussion, as shown in *Figure 2a*. They are DMN and CON, two higher order networks where shared peaks are higher count rank among shared peak occupied networks (the second-ranked and the third-ranked on macaque shared peaks; the fourth-ranked and the fifth-ranked on human shared peaks). The cingulo-opercular network (CON) is a brain network associated with action, goal, arousal, and pain. However, a study found that three newly discovered areas of the primary motor cortex that exhibit strong functional connectivity with the CON region, forming a novel network known as the somato-cognitive action network (SCAN; *Gordon et al., 2023*). The SCAN integrates body control (motor and autonomic) and action planning, consistent with the idea that aspects of higher level executive control might derive from movement coordination (*Llinás, 2002*; *Gordon et al., 2023*). CON may be shared in the form of the SCAN network across these two species. This could explain in part the results in *Figure 2a* that shared peaks are more on CONs. Default-mode network (DMN) is a ensemble of brain regions that are active in passive tasks, includes the anterior and posterior cingulate cortex, medial and lateral parietal cortex, and medial prefrontal cortex (*Buckner et al., 2008*). Although DMN is considered a higher order brain network, numerous studies have provided evidence of its homologous presence in both humans and macaques. Many existing studies have confirmed the similarity between the DMN regions in humans and macaques from various

perspectives, including cytoarchitectonic (*Parvizi et al., 2006*; *Buckner et al., 2008*; *Caminiti et al., 2010*) and anatomical tracing (*Vincent et al., 2007*). These studies all support the notion that some elements of the DMN may be conserved across primate species (*Mantini et al., 2011*). In general, the partial sharing of DMN between humans and macaques may be attributed to the higher occurrence of shared peaks within the DMN.

The consistency of peaks across individuals is an important indicator. It reflects the similarity in cortical folding morphology among individuals. When comparing the consistency of shared and unique peaks, it was discovered that shared peaks exhibit greater stability within the same species across different individuals. This observation was as expected because the similarities in folding patterns could be related to preferences for neurons to migrate in cortical areas (*Kriegstein et al., 2006*; *Friedrich et al., 2021*) and genetically coded (*Friedrich et al., 2021*). These genes, which regulate the cortical structural morphology, are likely conserved in both human and macaque brains over time. Therefore, this has led to the stable presence of cortical folding patterns in both human and macaque species. Moreover, it was found that the overall consistency of peaks in the macaque brain is much higher than that in the human brain, indicating that the spatial distribution of gyral peaks in the macaque brain is more concentrated across individuals compared to humans. This is possibly due to the simpler folding patterns that are more easily retained between individuals in macaque and human brains. Next, the correlation between shared peaks consistency across species was computed, revealing a positive correlation (not-significant though) between the consistency of human and macaque (*Figure 3*). This implies that peaks that are widespread in the human brain are also widespread in corresponding regions in the macaque brain. These findings further supports the homology of human and macaque brain structures (*Sereno and Tootell, 2005*; *Modha and Singh, 2010*).

The spatial distribution of shared/unique gyral peaks across species, as defined by our study, is not random but shows discernable patterns, which can be verified through statistical analysis of anatomical features (*Table 2*). The shared peaks, in comparison to the unique peaks, exhibit larger sulc and local surface area, but smaller curvature. Furthermore, in human-specific anatomical features (not available in the macaque dataset), the shared gyral peaks exhibit thinner cortex and greater myelination. The associations among these anatomical features can validate the regularity of the distribution of shared and unique peaks. The associations among these anatomical properties of the brain have been extensively verified in previous studies, including the strong positive correlation between sulc and local surface area (*Yang et al., 2016*), the negative correlation between myelin and curvature in most regions (*Schmitt et al., 2021*), and the negative correlation between local surface area and cortical thickness (*Maingault et al., 2016*). These findings confirm the validity of the anatomical characteristics of shared peaks and unique peaks. While many studies have confirmed the positive correlation of sulc and curvature throughout the whole brain (*Yang et al., 2016*), the sulc and curvature in our conclusion displayed opposite trends in both shared and unique clusters. Possible explanations to this are many folds: Firstly, all of the gyral peaks are defined within the gyri, and the correlation between sulc and curvature within the gyri is much weaker than that in the whole brain. In addition, the correlation between sulc and curvature in some areas is very low, such as the anterior cingulate (most are shared peaks), dorsolateral frontal cortex(most are unique peaks), and middle temporal gyrus (most are unique peaks; *Schmitt et al., 2021*). This non-uniform spatial distribution leads to the disappearance of the correlation between sulc and curvature. Therefore, the anatomical patterns within the peaks and the global patterns of the entire brain are not in conflict.

Through evaluating the structural and functional connectivity properties of shared and unique peaks, it was observed that shared peaks exhibit larger connectivity attributes, such as degree, strength, clustering coefficient, betweeness and efficiency, compared to unique peaks. Higher degree and strength values suggest that shared peaks are connected to more vertices in the brain network. Additionally, it was found that clustering coefficient and efficiency, which measure local information transmission capacity and resilience to random attacks in a network, were higher in shared peaks. Betweeness, a centrality measure that quantifies the importance of a node in the network, also showed higher values for shared peaks, indicating greater importance of these peaks in the brain network. These results suggest that shared peaks may play a role as network hubs in contrast to unique peaks. Gyral peaks exhibit a high degree of connectivity within local neighborhoods, creating a 'small world' structure within the network, and may behave as hubs in the structural/functional network, as suggested by previous studies (*Sporns and Zwi, 2004*; *Bassett and Bullmore, 2006*; *Bullmore and Sporns, 2009*;

*He and Evans, 2010*). In many studies, higher order brain regions like the DMN are recognized as the global network hubs and the communication centers of the brain's global network. These regions typically exhibit higher node degree and strength. However, there is an interesting finding in our study. In the human brain, the more shared peaks (about 65%) are located in lower order brain regions, while unique peaks are mainly (about 74%) located in higher order regions. However, this trend is relatively less pronounced in the macaque brain. There are two possible explanations for this. Firstly, peaks is defined at a much more local scale, in contrast to the definition of brain functional regions, such as DMN. This seemingly contradictory findings could be reconciled by their definitions of 'network hubs' at respective coarse and fine scales. Specifically, while higher order brain regions such as DMN serve as the information exchange centers for large-scale brain network, the information transfer within each region at a finer scale could be primarily facilitated by loci, such as the shared peak. These findings suggest that, peaks that are in larger-scale DMN while exhibiting lower hub-like attributes at a vertex-level, could be referred to as provincial hubs (*Guimerà and Nunes Amaral, 2005*; *Hwang et al., 2017*). This can be understood as the preservation of the most fundamental and mainstream topological structure and communication patterns during the evolutionary process of species, while species-specific peaks that appear later in the evolutionary process may serve higher order and more specific functions (*Goulas et al., 2014*; *Rilling, 2006*). Another issue worth discussing is the relationship between degree and clustering coefficient. Some studies focusing on social networks and random intersection graph models have found that clustering coefficient correlates negatively with degree *Foudalis et al., 2011*; *Bloznelis, 2013*. While in our study, when comparing the functional network characteristics of shared and unique peaks, it was found that the patterns of degree and clustering coefficient were similar (3). The differences in network characteristics between brain networks and social networks or random networks may reflect distinct organizational patterns in the brain compared to other networks. Furthermore, due to our focus on the internal properties of peaks in this study, the patterns observed may not align entirely with the principles followed by the entire brain network.

Through comparisons with multiple brain atlases, it was observed that there are more diverse brain regions around shared peaks than around unique peaks for multiple brain atlases with a median parcellation resolution. It is noted that the observation is not consistent on atlases with relatively lower resolutions (e.g. BA05 for macaque, n=30 and Yeo2011 for human, n=7) or, in particular, higher resolutions (e.g. n≥500 for Schaefer-500 and n≥200 for Vosdewael-400). This inconsistency is reasonable since the resolution of the parcellation itself will largely determines the chance of a cortical region appear in a peak's neighborhood, if the parcellation resolution is too coarse or too fine. For example, if n=1 (the entire cortex is the only one region) or n=30 k (each vertex is a region), each peak will has the same number of neighboring regions for these two extreme cases (one brain region for each peak for n=1; around 30 vertices for each peak for n=30 k). This finding may suggest a higher diversity of brain functions associated with shared peaks. From a microscopic perspective, brain function is determined by the structure and functional characteristics of cells. The brain is composed of various types of cells, and each type of cell contributes to different aspects of brain function. The differential expansion of cortical regions and the introduction of new functional modules during the process of evolution may be the result of changes in progenitor cells (*Clowry et al., 2018*). In this experiment, the shared peaks represent regions with less cortical expansion, indicating a smaller proportion of ancestral cells. It may allow them to participate in a greater variety of brain functions and be surrounded by more diverse brain regions. From a macroscopic perspective, in the analysis of brain folding, a traditional approach is to partition the brain into a set of distinct regions, known as parcellation, based on functional, structural, or cytoarchitectural criteria. This parcellation serves as the most common unit of analysis in studying brain folds. This well-defined partitioning method provides an intuitive framework for analyzing the brain, leading to computational, statistical, and interpretational efficiencies *Eickhoff et al., 2018*; *Glasser et al., 2016*. Simply averaging all vertex characteristics within a region assumes the homogeneity within the region and only one dominant pattern *Haak and Beckmann, 2020*. However, both functional and microstructural properties often highly variable within a region, and inconsistent across modalities. Additionally, adjacent vertices in different regions may also have similar characteristics. Therefore, boundaries vary depending on the chosen modality, and no clear boundaries are evident in all modalities or analysis approaches. The brain has no true 'boundaries'. In this study, it was observed that shared peaks in regions surrounded by a larger number of neighboring brain regions are more likely to be assigned to the 'boundaries' of those regions across different

classification approaches. Therefore, it is speculated that these shared peaks might be involved in a more diverse range of brain functions.

Using lasso regression, 28 genes were screened on the cortex, identifying significant contributions to the classification of shared and unique peaks. Further applying Welch's t-test, significant differential expression was found in seven genes between the shared and unique peak regions. Among them, SNAP29 and KCNH5 are closely associated with neuronal activity and brain function, and these two genes show higher and lower expression levels in the shared peaks, respectively. While, low expression of SNAP29 protein levels disrupts neural circuits in a presynaptic manner, leading to behavioral dysfunctions *Yan et al., 2021*. Therefore, the majority of shared peaks located in lower-level brain regions exhibit higher SNAP29 expression, aiming to minimize the occurrence of low SNAP29 expression that could disrupt neural circuits and result in behavioral dysfunctions. Another differentially expressed gene was KCNH5. The voltage-gated Kv10.2 potassium channel, encoded by KCNH5, is broadly expressed in mammalian tissues, including the brain. According to previous studies, dysfunction of Kv10.2 may be associated with epileptic encephalopathies and autism spectrum disorder (ASD) (*Hu et al., 2022*). And these two diseases happen to be more prevalent in humans, coinciding with the high expression of the KCNH5 gene in unique peaks.

# Materials and methods
## Dataset description
### Human MRI
In this study, the Human Connectome Project (HCP) S900 Subjects MR imaging data from Q3 Release (https://www.humanconnectome.org/). The data was obtained from the Q3 Release and all participants involved provided written informed consent and the study was approved by the relevant institutional review boards. The MR images were acquired by a Siemens 'Connectome Skyra' 3T scanner housed at Washington University in St Louis using a 32-channel head coil. For T1-weighted MRI: TR = 2400 $ms$, TE = 2.14 $ms$, flip angle = 8 $deg$, FOV = 224× 224 $mm$ and resolution = 0.7×0.7×0.7 $mm^3$. T2-weighted MRI: TR = 3200 $ms$, TE = 565 $ms$ and resolution = 0.7×0.7×0.7 $mm^3$. Diffusion MRI (dMRI): TR = 5520 $ms$, TE = 89.5 $ms$, refocusing flip angle = 160 $deg$, flip angle = 78 $deg$, FOV = 210×180 $mm$, matrix = 168×144, resolution = 1.25×1.25×1.25 $mm^3$, 1.25 $mm$ isotropic voxels, echo spacing = 0.78 $ms$, BW = 1,488 $Hz/Px$. Resting state fMRI (rfMRI): TR = 720 $ms$, TE = 33.1 $ms$, flip angle = 52 $deg$, FOV = 208×180 $mm$, matrix = 104×90, 1200 time points, 2.0 $mm$ isotropic voxels, BW = 2,290 $Hz/Px$.

The standard HCP MR structural pipelines (*Glasser et al., 2013*; *Fischl, 2012*; *Jenkinson et al., 2002*; *Jenkinson et al., 2012*) were applied for processing all structural MR images. It mainly includes the following three main steps: (1) PreFreeSurfer pipeline (*Jovicich et al., 2006*; *van der Kouwe et al., 2008*; *Smith, 2002*) which corrected for image distortion, aligned and averaged T1w and T2w images and registered the subject's native structural volume space to MNI space. (2) FreeSurfer pipeline (*Dale et al., 1999*; *Fischl et al., 2002*; *Fischl et al., 1999*; *Ségonne et al., 2005*) including segmentation of brain volume, reconstruction of white matter and pial surfaces, and registering to fsaverage surface atlas; (3) PostFreeSurfer pipeline, including surface registration to the Conte69 surface template (*Van Essen et al., 2012b*) by using MSM-All algorithm (*Glasser et al., 2016*; *Robinson et al., 2018*; *Robinson et al., 2014*). In this step, cortical folding, myelin maps, and resting state fMRI (rfMRI) correlations together for registration, which improved the cortical correspondences across different subjects. For this study, the white matter cortical surface with 64,984 vertices after MSM-All registration and the associated cortical folding features such as sulc, myelin, and cortical thickness, were adopted for cross-subjects analysis. For the diffusion MRI (dMRI) data, fiber tractography was performed using MRtrix3 (*Tournier et al., 2019*, https://www.mrtrix.org). Each individual had 40,000 fiber tracts reconstructed. A maximum length limit of 150 mm was defined to reduce the presence of false positives (*Varriano et al., 2018*).

### Gene expression data
The AHBA microarray gene expression data consists of 3702 samples from six typical adult human brains. Several hundred samples (mean ± standard deviation: 617±241) were collected from cortical, subcortical, brainstem and cerebellar regions in each brain to profile genome-wide gene expression. In the AHBA, each gene probe is associated with a numerical ID and a platform-specific label or

name. If a probe is assigned to represent a unique gene it is also characterized with a range of gene-specific labels such as gene symbol and an Entrez Gene ID–a stable identifier for a gene generated by the Entrez Gene database at the National Center for Biotechnology Information (NCBI). The probe-level data offer high-resolution coverage of nearly the entire brain, providing expression measures for over 20,000 genes from 3702 spatially distinct tissue samples. The AHBA data is available at figshare https://figshare.com/s/441295fe494375aa0c13. The AHBA dataset has been preprocessed, and detailed information can be referred to *Arnatkeviciute et al., 2019*. The first six processing steps produce the region×gene matrix that can be used for the regional analyses.

## Macaques MRI

Rhesus macaque monkeys' structural and functional MR imaging data aging from 0.8 to 4.5 years were selected from the non-human primate (NHP) consortium PRIME-DE from University of Wisconsin–Madison (http://fcon_1000.projects.nitrc.org/indi/indiPRIME.html). The full dataset consisted of 592 rhesus macaque monkeys (*Macaca mulatta*) scanned on a 3T with a 4-channel coil. For T1-weighted MRI: TR = 11.4 $ms$, TE = 5.41 $ms$, flip angle = 10 $deg$, image matrix = 512×248×512 and resolution = 0.27× 0.50×0.27 $mm^3$. The rsfMRI data were preprocessed based on DPARSF, which included slice timing, realignment, covariant regression, band-pass filtering (0.01–0.1 $Hz$), and smoothing (FWHM = 4 $mm$). T1w images were fed into CIVET, registering them into the NMT-standardized space (*Seidlitz et al., 2018*) using an affine transformation, followed by image resampling and tissue segmentation. The reconstructed white matter cortical surface was obtained using Freesurfer. The surfaces were resampled to 40 k vertices to ensure vertex-to-vertex correspondence across subjects through spherical registration. After linear registration between fMRI and T1w MRI via FLIRT, the volume time-series were mapped to surface vertices for further analysis.

## Peak cluster extraction

Based on our previous work, gyral peaks are defined as the highest point of the gyri (*Zhang et al., 2022*). Gyral height was measured by 'Sulc' (https://surfer.nmr.mgh.harvard.edu/, *Fischl, 2012*), which was defined as the displacement from a vertex on the surface to a hypothetical mid-surface, which is between the gyri and sulci, and the 'mean' of displacements of all vertices is zero (*Fischl et al., 1999*). Thus, gyral peaks on individuals were identified by locating the vertex of the minimum sulc value within the x-ring (4-ring for humans, 3-ring for macaques) neighborhood on the grid (*Zhang et al., 2022*; *Zhang et al., 2023*). To obtain group-wise peak clusters, all gyral peaks in individual spaces of the two species were projected onto the respective template white matter surface, which produced a count map of peaks for each species. Of note, vertex-to-vertex correspondences were established across all surfaces within each species. Next, peak count maps of two species were processed by anisotropic smoothing, with n iterations within an k-ring neighborhood, as described in *Meng et al., 2014*; *Zhang et al., 2023*. Finally, the watershed clustering algorithm detailed in *Meng et al., 2014*; *Rettmann et al., 2002*; *Yang and Kruggel, 2008*; *Zhang et al., 2023* was applied to the smoothed count map to automatically generate group-wise peak clusters for each species. Notably, the selection of parameters for anisotropic smoothing and watershed clustering algorithm were based on the previous work (*Zhang et al., 2022*; *Zhang et al., 2023*). Parameters of these three steps (individual peak extraction, anisotropic smoothing and watershed clustering algorithm) on two species are reported in the Supplementary Information. In total, 192 (LH: 96, RH: 96) and 85 (LH: 42, RH: 43) peak clusters were detected on Humans and macaques, respectively (*Figure 1a*).

## Cross-species registration

To elucidate the inter-species relationship of group gyral peaks between humans and macaques, a functional joint alignment technique (*Xu et al., 2020*) was employed to project macaque peak clusters onto the human cortical surface. They first constructed a joint similarity matrix by concatenating within- and cross-species similarities of connectivity patterns. Next, the diffusion embedding algorithm applied on the similarity matrix. Finally, gradients as surface features, and the cortical surfaces of humans and macaques were aligned using Multimodal Surface Matching (MSM) (*Robinson et al., 2014*). This technique builds upon recent advances in high-dimensional common space representations of functional organization and offers a transformational framework between human and macaque cortices.

### Definition of shared and unique peak clusters

After the cross-species registration mentioned above, the group-wise gyral peak clusters of the two species were placed on the same template surface. The determination of peak clusters that are shared between species involves two criteria: (1) the Dice of clusters >0; and (2) the geodesic distance between the centers of the two clusters is less than 7 *mm*. If a pair of clusters satisfies either one of these two criteria, they can be identified as peak clusters that are shared between species. The difference set between all peaks of the two species and the shared peaks is the set of unique peaks for each species.

### Statistical analysis

All variables used in the two-samples t-test follow a normal distribution and all p-values were corrected for multiple comparisons using the false discovery rate (FDR) method. Moreover, in order to identify differently expressed genes within shared and unique peaks, and considering the unequal sample sizes for shared and unique peaks, the Welch's t-test was employed, which is suitable for this scenario. For all tests, a p-value <0.05 was considered significant (FDR corrected).

### Anatomical features of gyral peaks

The anatomical characteristics of shared and unique gyral peaks were analyzed, including sulc, curvature (the amount of bending at a point on a convoluted surface), thickness (the distance from the point on the pial surface to the nearest point on the white surface *Fischl and Dale, 2000*) myelin, and local surface area (The calculation that the average area of all triangles in the neighborhood of vertex $i$ is $S_i$. The local surface area of vertex $i$ is the mean neighborhood area $S_i$ divided by the mean of all vertices in the whole brain $\bar{S}$).

### Functional and structural connectivity

The white matter surface (excluding the regions between two hemispheres) was parcellated into 1400 patches for human and 1700 patches for macaque (*He et al., 2022*) due to the number of vertices on the surface. A structural connective graph $G_s = \{V, E_s, A_s\}$ and a functional connective graph $G_f = \{V, E_f, A_f\}$ were constructed for each subject. Graph nodes $v_s$ and $v_f$ were defined as cortical patches of the same area. For human individual structural connectivity matrix $A_s$, $a_s^{ij}$ represents the fiber count connecting the two nodes. For human and macaque individual functional connectivity matrices, the Pearson correlation coefficient (PCC) between the average time-series between two nodes $v_f^i$ and $v_f^j$ was calculated, followed by Fisher's z-transformation. Due to the vertex-to-vertex correspondences across individual surfaces of each species, the patches (or nodes) had cross-subject correspondences as well. On this basis, the structural and functional connectivity matrices of each subject were averaged to obtain a group-average structural and functional connectivity matrix $\bar{A}_s$ and $\bar{A}_f$. Then, for each row in the group-average functional connectivity matrix, the values of the top 10% of connections were retained, whereas all others were zeroed. On this group-average graph, nodal graph metrics, including degree, strength, clustering coefficient, betweeness, and efficiency, were computed using the Brain Connectome Toolkit (https://sites.google.com/site/bctnet/). The definitions of these network properties are detailed in the Supplementary Information.

### Feature selection of genes

Since human gyral peaks were divided into peaks shared with macaque and peaks unique to human, the aim was to investigate the genes that are significantly different expressed between two types of gyral peak. The preprocessed AHBA gene data is in the form of region×gene and the region above referred to the parcellation of a certain atlas, such as Aparc, Schaefer100, Schaefer500, Schaefer1000, etc. The Schaefer500 atlas was selected for this study because high resolution may result in some areas with no gene data (more details refer to *Arnatkeviciute et al., 2019*), while low resolution may result in multiple categories of clusters being located in the same region. Therefore, Schaefer500 was chosen as the most suitable atlas for this work. First, all regions of Schaefer500 atlas were labeled as shared, unique, or other based on the positions of group-wise gyral peaks. Then, Lasso (a linear regression method that uses L1 regularization for gene selection) was applied on this labeled gene data. The cost function of Lasso regression is as follows: $Cost(w) = \sum_{i=1}^{N}(y_i - w^T x_i)^2 + \lambda \|w\|_1$. An important parameter of Lasso is lambda, which affects the sparsity of feature selection. Ten-fold cross-validation

was employed to select the optimal lambda. By considering the maximization of accuracy (acc) and minimization of mean squared error (MSE) simultaneously, the lambda value was ultimately determined to be 0.027 (*Figure 5b*). The accuracy of training set was 0.84, and the MSE was 0.64; The accuracy of test set was 0.75, and the MSE was 1.00.

## Data availability

All human data analyzed in this manuscript were obtained from the open-access HCP adult sample (https://www.humanconnectome.org/). Macaque data came from PRIME-DE (http://fcon_1000.projects.nitrc.org/indi/indiPRIME.html). Fiber tracking based on MRtrix3 (https://www.mrtrix.org). Full extraction process of gyral peaks of this work can be found at https://github.com/zsy0728/extract-gyral-peak (copy archived at *Zhang, 2024*).

## Acknowledgements

The authors express gratitude to the various contributors to the open access databases that our data was downloaded from. HCP data were provided by the Human Connectome Project, WU-Minn Consortium (Principal Investigators: David Van Essen and Kamil Ugurbil; 1U54MH091657) funded by the 16 NIH Institutes and Centers that support the NIH Blueprint for Neuroscience Research; and by the McDonnell Center for Systems Neuroscience at Washington University. Macaque data were provided by the PRIME-DE. Primary support for the work by Michael P Milham and the INDI team was provided by gifts from Joseph P Healy to the Child Mind Institute, as well as by the BRAIN Initiative (R01MH111439). MPM is a Randolph Cowen and Phyllis Green Scholar. Primary support for the work by Charles Schroeder is provided by the BRAIN Initiative (R01MH111439) and the Sylvio O Conte Center 'Neurobiology and Dynamics of Active Sensing' (P50MH109429). Primary support for the work by Daniel Margulies is provided by the Max Planck Society.

## Additional information

### Funding

| Funder | Grant reference number | Author |
| --- | --- | --- |
| National Natural Science Foundation of China | 31971288 | Tuo Zhang |
| Key Program of the National Natural Science Foundation of China | 62131009 | Tuo Zhang |
| Innovation Foundation for Doctor Dissertation of Northwestern Polytechnical University | CX2022053 | Songyao Zhang |
| Innovation Foundation for Doctor Dissertation of Northwestern Polytechnical University | CX2022052 | Zhibin He |
| National Key Research and Development Program of China | 2020AAA0105701 | Junwei Han |
| National Natural Science Foundation of China | 61936007 | Junwei Han Lei Guo |
| National Natural Science Foundation of China | U20B2065 | Junwei Han |
| National Natural Science Foundation of China | U1801265 | Junwei Han |

The funders had no role in study design, data collection and interpretation, or the decision to submit the work for publication.

## Author contributions
Songyao Zhang, Conceptualization, Formal analysis, Validation, Visualization, Methodology, Writing - original draft; Tuo Zhang, Conceptualization, Writing - review and editing; Guannan Cao, Tao Liu, Validation; Jingchao Zhou, Data curation; Zhibin He, Formal analysis; Xiao Li, Methodology; Yudan Ren, Investigation; Xi Jiang, Conceptualization; Lei Guo, Junwei Han, Funding acquisition; Tianming Liu, Project administration

## Author ORCIDs
Songyao Zhang ⓘ http://orcid.org/0000-0002-5406-3643
Tuo Zhang ⓘ http://orcid.org/0000-0002-6075-3384
Zhibin He ⓘ http://orcid.org/0000-0002-5604-1886

Reviewer #1 (Public Review): https://doi.org/10.7554/eLife.90182.3.sa1
Author response https://doi.org/10.7554/eLife.90182.3.sa2

---

# Additional files

## Supplementary files
• MDAR checklist

## Data availability
The data utilized in this manuscript were obtained from three distinct sources. Human MRI data analyzed in this study were sourced from the open-access Human Connectome Project (HCP) adult sample, available at https://www.humanconnectome.org/. Macaque data were obtained from the PRIME-DE database, accessible at http://fcon_1000.projects.nitrc.org/indi/indiPRIME.html. The Allen Human Brain Atlas (AHBA) gene dataset, after preprocessing, is available for download at https://figshare.com/s/441295fe494375aa0c13. All data are publicly available and can be accessed through the provided links.

The following previously published dataset was used:

| Author(s) | Year | Dataset title | Dataset URL | Database and Identifier |
|---|---|---|---|---|
| Arnatkeviciute A, Fulcher BD, Fornito A | 2020 | AHBA data | https://figshare.com/s/441295fe494375aa0c13 | Figshare, s/441295fe494375aa0c13 |

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

# Appendix 1

## Parameter selection

*Appendix 1—table 1* presents all the parameters used in the three algorithms for detecting individual and group peaks in the two species, along with their corresponding meanings. X-order ring neighbor was utilized to detect peaks on individual surfaces and chose k-order ring plus n-iterations for anisotropic smooth algorithm for the count map. Parameters for watershed clustering algorithm are related to the value of the count map. fg and bg respectively determine the minimum and maximum count of the segmented area. A smaller value of parameter merge results in more clusters. All parameters were determined based on previous studies (*Zhang et al., 2022*; *Zhang et al., 2023*).

**Appendix 1—table 1.** Parameter selection of gyral peaks detection in human and macaque.

| Assignments | Parameters | Meanings | Human | Macaque |
|---|---|---|---|---|
| Detect Individual Peaks | x | Search peaks ring | 4 | 3 |
| | k | Anisotropic Smooth ring | 1 | 1 |
| Anisotropic Smooth | n | Smooth iteration | 20 | 20 |
| | fg | Minimum count value for cluster coverage | 45 | 4 |
| | bg | Maximum count value for cluster coverage | 209 | 28 |
| Watershed Clustering | merge | Determinants of whether two clusters are merged | 7 | 3 |

# Appendix 2

## Locations of Peaks in Human and Macaque

*Appendix 2—tables 1 and 2* displays the locations of all peaks in the human and macaque brain. The region names are derived from the aparc2009 atlas of human and BA05 atlas of macaque.

**Appendix 2—table 1.** The location of human peak clusters.

| Location | Human Cluster Number | Location | Human Cluster Number |
|---|---|---|---|
| G_and_S_frontomargin | 61 | G_oc-temp_med-Parahip | 3,9,15,63,74,108 |
| G_and_S_occipital_inf | 120,130 | G_orbital | 20,25,52,55,60,62,64,69,92,113 |
| G_and_S_paracentral | 2,28 | G_pariet_inf-Angular | 175,189 |
| G_and_S_subcentral | 23,38 | G_pariet_inf-Supramar | 156,171,172 |
| G_and_S_transv_frontopol | 51,182 | G_parietal_sup | 140,178,185,187 |
| G_and_S_cingul-Ant | 40,44,89,105,115,162,186 | G_postcentral | 33,35,99,109,124,139,153,170 |
| G_and_S_cingul-Mid-Ant | 49,70,72,158 | G_precentral | 18,39,53,65,66,67,87,90,107 |
| G_and_S_cingul-Mid-Post | 45,79 | G_precuneus | 88,100,117,143,180,192 |
| G_cingul-Post-dorsal | 26,37,78,98,102 | G_rectus | 7,13,22,27,41,82 |
| G_cingul-Post-ventral | 29,150,154 | G_temp_sup-Lateral | 47,59,68,81,93,114,116 |
| G_cuneus | 21,31,34,36 | G_temporal_inf | 95,123,128,160 |
| G_front_inf-Opercular | 42,43,125 | G_temporal_middle | 76,104,112,129,173 |
| G_front_inf-Orbital | 80 | Pole_occipital | 1,11,17,24,57,142 |
| G_front_inf-Triangul | 132,135,138 | Pole_temporal | 6,14,106,141,146 |
| G_front_middle | 131,149,152,165,179,184,188 | S_calcarine | 83 |
| G_front_sup | 46,71,73,75,77,84,85,91,97,118,126,137,157,161,174,181,183,191 | S_front_middle | 190 |
| G_Ins_lg_and_S_cent_ins | 16,32,54,119 | S_front_sup | 166,169 |
| G_insular_short | 4,12 | S_intrapariet_and_P_trans | 159 |
| G_occipital_middle | 101,110,121,127,155 | S_oc-temp_med_and_Lingual | 151 |
| G_occipital_sup | 96,103,144,145 | S_orbital-H_Shaped | 122 |
| G_oc-temp_lat-fusifor | 48,58,147,163,164,167 | S_pericallosal | 50,94 |
| G_oc-temp_med-Lingual | 8,10,19,56,133,134,136,148,168,176,177 | S_subparietal | 111 |

**Appendix 2—table 2.** The location of macaque peak clusters.

| Location | Macaque Cluster Number | Location | Macaque Cluster Number |
|---|---|---|---|
| Area 2 | 11,15,37,43,48 | Area 13 | 5,7,10,13,16,46,49,54,77 |
| Area 3 | 4,32,33,69,74 | Area 14 | 31,34,50,55,83 |
| Area 4 | 9,27,56,70,71 | Area 15 | 25,26 |
| Area 5 | 1,44 | Area 16 | 14,42,79,85 |
| Area 6 | 22,35,58,80 | Area 20 | 3,18,38,39,41,47,51,57,61,62,63 |
| Area 7 | 2,6,12,17,28,45,53,60,65,67,76,82 | Area 21 | 8,23,24,30,40,64,66,81 |
| Area 9 | 19,29,68,72 | Area 23 | 75 |
| Area 12 | 20,21,52,59,73,84 | Area 24 | 36,78 |

## Appendix 3

### Locations of Shared and Unique Peak Clusters

*Appendix 3—figure 1* shows the locations of shared peak clusters of macaque on the surface of the macaque brain template. The locations of all human shared peaks are reported in *Appendix 3—table 1*.

# Shared Peak Clusters of Macaque

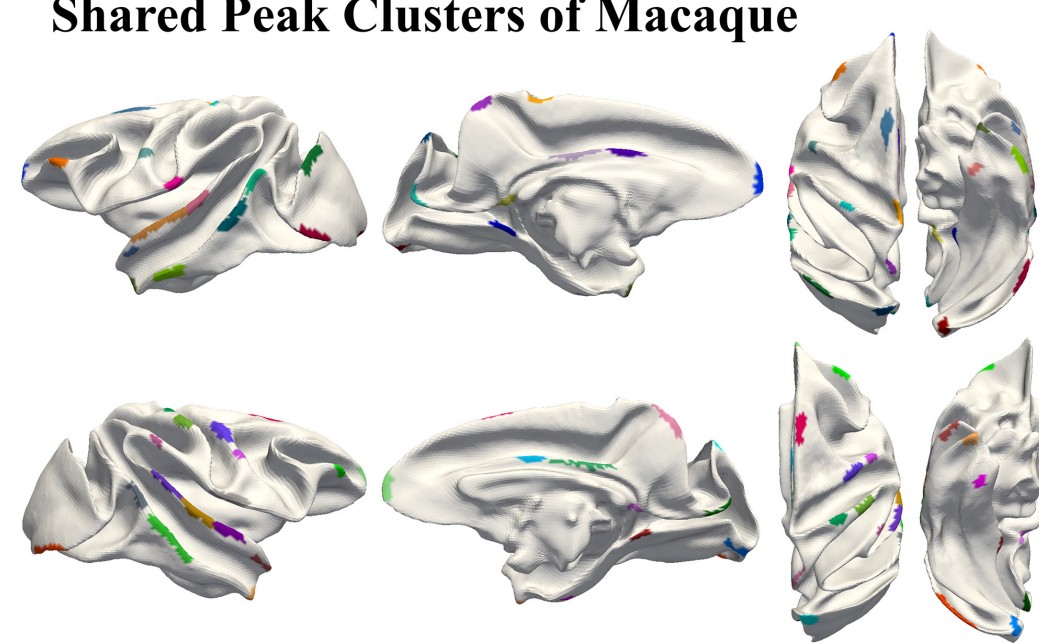

Views:       Lateral              Medial              Dorsal-Ventral

**Appendix 3—figure 1.** Macaques share peak clusters display on the surface of the macaque brain template.

The main text presents the counts of peak cluster centers occurrence in different networks (Cole-Anticevic). To mitigate the influence of different brain area sizes on the count of clusters, the count was normalized by the regional surface area and reported in *Appendix 3—figure 2*. The results were similar to those on the original counts. These findings indicated that most of the shared peaks are located in low-order sensory and motor networks, while most of the unique peaks are located in higher-order networks. The regions with the highest density of shared peak cluster centers are V1, Aud, and VMN, while the region with the highest density of unique peak cluster centers is Lan, DMN, OAN, and FPN. Further explanations of some of the different observations in *Figure 2* and *Appendix 3—figure 2* can be found in the 'Discussion and Conclusion' section.

**Appendix 3—table 1.** Location of shared peak clusters on human.

| Clusters in LH | Location | Clusters in RH | Location |
|---|---|---|---|
| LH Shared 1 | G_front_sup | RH Shared 1 | G_and_S_transv_frontopol |
| LH Shared 2 | G_and_S_subcentral | RH Shared 2 | G_postcentral |
| LH Shared 3 | G_cuneus | RH Shared 3 | G_temp_sup-Lateral |
| LH Shared 4 | G_oc-temp_med-Parahip | RH Shared 4 | G_occipital_sup |
| LH Shared 5 | G_temp_sup-Lateral | RH Shared 5 | G_orbital |
| LH Shared 6 | G_occipital_middle | RH Shared 6 | Pole_temporal |
| LH Shared 7 | G_precentral | RH Shared 7 | Pole_occipital |
| LH Shared 8 | G_temp_sup-Lateral | RH Shared 8 | G_front_inf-Opercular |

*Appendix 3—table 1 Continued on next page*

*Appendix 3—table 1 Continued*

| Clusters in LH | Location | Clusters in RH | Location |
|---|---|---|---|
| LH Shared 9 | G_orbital | RH Shared 9 | G_temp_sup-Lateral |
| LH Shared 10 | G_postcentral | RH Shared 10 | G_oc-temp_med-Parahip |
| LH Shared 11 | Pole_temporal | RH Shared 11 | G_precentral |
| LH Shared 12 | G_and_S_cingul-Mid-Ant | RH Shared 12 | Pole_occipital |
| LH Shared 13 | G_oc-temp_med-Lingual | RH Shared 13 | G_occipital_middle |
| LH Shared 14 | G_parietal_sup | RH Shared 14 | G_postcentral |
| LH Shared 15 | G_oc-temp_med-Lingual | RH Shared 15 | Pole_occipital |
| LH Shared 16 | Pole_occipital | RH Shared 16 | G_precuneus |
| LH Shared 17 | G_oc-temp_med-Parahip | RH Shared 17 | G_precentral |
| LH Shared 18 | G_and_S_occipital_inf | RH Shared 18 | G_pariet_inf-Supramar |
| LH Shared 19 | S_front_sup | RH Shared 19 | G_front_sup |
| LH Shared 20 | Pole_temporal | RH Shared 20 | Unkown |
| LH Shared 21 | G_precentral | RH Shared 21 | G_temp_sup-Lateral |
| LH Shared 22 | G_cingul-Post-ventral | RH Shared 22 | G_cingul-Post-ventral |
| LH Shared 23 | Pole_occipital | RH Shared 23 | G_and_S_cingul-Mid-Ant |
| LH Shared 24 | Pole_occipital | RH Shared 24 | G_oc-temp_lat-fusifor |
| LH Shared 25 | G_and_S_cingul-Mid-Ant | RH Shared 25 | G_oc-temp_med-Lingual |
| | | RH Shared 26 | G_and_S_cingul-Mid-Post |

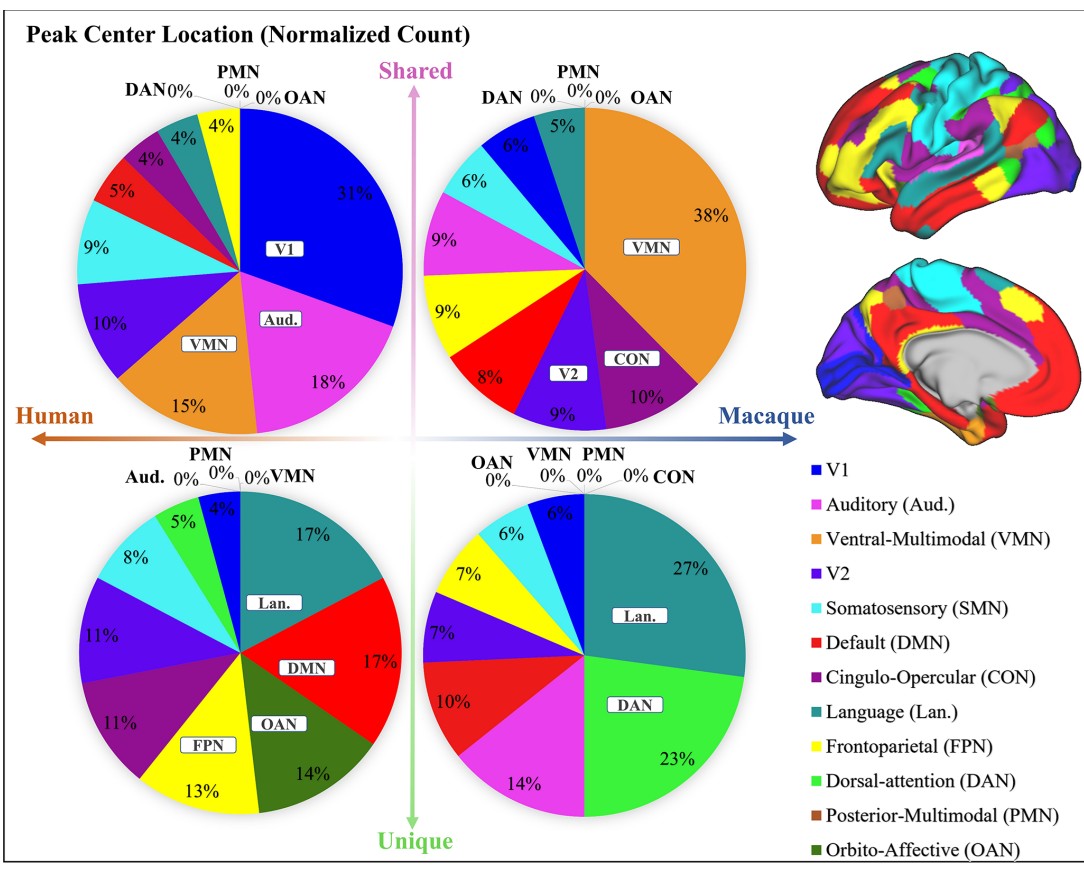

**Appendix 3—figure 2.** Pie chart shows the normalized count of shared and unique peaks across different brain networks both for human and macaque. Right panel shows the Cole-Anticevic (CA) networks (*Ji et al., 2019*) on human surface as a reference.

## Appendix 4

### Confidence of Shared Peaks

*Appendix 4—figure 1a* illustrates the locations of all shared peaks. There are two definitions of shared peaks: (1) the Dice of clusters >0; and (2) the geodesic distance between the centers of the two clusters is less than 7mm. The credibility of shared peak clusters defined by the coincidence rate of clusters is measured using the overlap ratio. The higher the overlap rate of clusters, the higher the confidence of shared clusters between species *Appendix 4—figure 1b*. The credibility of shared peak clusters defined by the distance of cluster centers is measured using the ranking of center distances. The distance between the two closest clusters in the whole brain is set as 100 points, and the distances of other clusters are proportionally reduced accordingly *Appendix 4—figure 1c*.

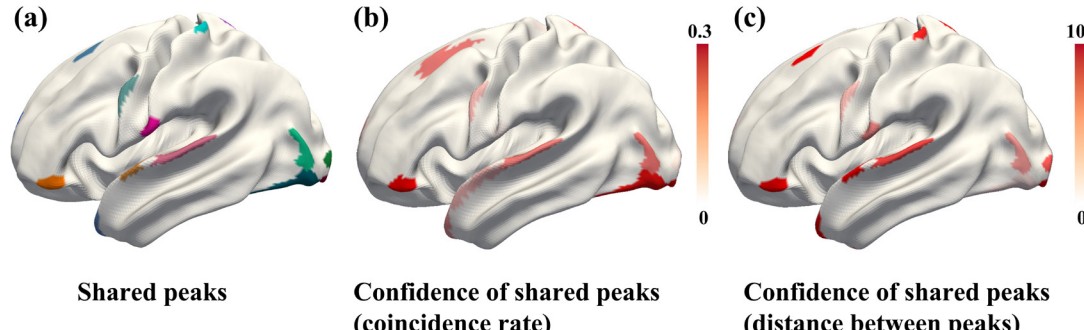

**(a)** Shared peaks

**(b)** Confidence of shared peaks (coincidence rate)

**(c)** Confidence of shared peaks (distance between peaks)

**Appendix 4—figure 1.** Confidence of shared peak clusters. (**a**) Location of shared peaks. (**b**) Confidence of shared peak clusters defined by the coincidence rate of clusters between human and macaque. (**c**) Confidence of shared peak clusters defined by the distance of cluster centers between human and macaque.

## Appendix 5

## Network properties for graph analysis

### Network Properties

Degree: Degree is the most important description of the statistical characteristics of node connection. Degree $K_i$ is defined as the number of edges directly connected to a node i. The greater the degree of the node, the more connections the node has, and the more important the status of the node in the network. G is an undirected weighted network in our work, the degree of node i is defined as:

$$K_i = \sum\nolimits_{j=1}^{N} a_{ij} \tag{1}$$

Strength: For a N-node weighted network G which weight matrix is W, the strength of node i is defined as:

$$S_i = \sum\nolimits_{j=1}^{N} w_{ij} \tag{2}$$

Cluster Coefficient: The clustering coefficient of a vertex i is the probability that the neighbours of this vertex (all other vertices to which it is connected by an edge) are also connected to each other. The clustering coefficient of a vertex ranges between 0 and 1.

$$C_i = \frac{2e_i}{k_i(k_i - 1)} = \frac{\sum_{j,m} a_{ij} a_{im} a_{mj}}{k_i(k_i - 1)} \tag{3}$$

Betweenness Centrality: Betweenness Centrality indicates the times of a node appears on all shortest paths in a graph. $\sigma_{st}$ is the number of shortest paths from node $s$ to node $t$, and $\sigma_{st}(v_i)$ is the number of times those paths pass through $(v_i)$.

$$BC_i = \sum_{s \neq i \neq t} \frac{\sigma_{st}(v_i)}{\sigma_{st}} \tag{4}$$

Efficiency: The average communication efficiency of the network G is then defined as the average over the pairwise efficiencies:

$$E(i) = \frac{1}{N_{G_i}(N_{G_i} - 1)} \sum_{j \neq k \in G_i} \frac{1}{l_{j,k}}, \tag{5}$$

# Appendix 6

## Structural connectivity

**Appendix 6—table 1.** The mean (± SD) structural connectivity characteristics of shared and unique peak clusters of human.

The bold font represent the larger values between the shared peak and unique peaks. *indicates $P<0.05$; **indicates $P<0.01$,***indicates $P<0.001$

|  | Degree | Strength | CC | Betweeness | Efficiency |
|---|---|---|---|---|---|
| Shared | 31.37±5.04 | 32.29±5.79 | 0.23±0.04 | 5.50±2.12(×10³) | 0.43±0.05 |
| Unique | 31.79±3.23 | 29.46±3.43 | 0.19±0.02 | 5.10±1.08(×10³) | 0.39±0.03 |
| p | <0.01 | <0.001 | <0.001 | <0.001 | <0.001 |
| t | 2.39 | 9.08 | 23.01 | 5.66 | 21.20 |

# Appendix 7

## Spatial Relationship Between Peaks and Functional Regions

A statistical analysis of the number of functional brain regions appearing in the neighborhoods of shared and unique peaks was conducted. The outcomes, derived from the utilization of comprehensive human brain atlases, were summarized in *Appendix 7—table 1*.

**Appendix 7—table 1.** The mean values (± SD) of brain regions where shared and unique peaks appeared within a 3-ring neighborhood in 21 common human atlases.
The p-values were corrected by FDR.

| Atlas Name | Yeo2011(7) | Glasser2016 | Schaefer-100 | Schaefer-200 | Schaefer-300 | Schaefer-400 | Schaefer-500 |
|---|---|---|---|---|---|---|---|
| Share Nbr | 1.48±0.10 | 2.43±0.15 | 1.89±0.12 | 2.12±0.11 | 2.23±0.11 | 2.46±0.13 | 2.50±0.14 |
| Unique Nbr | 1.54±0.07 | 2.37±0.09 | 1.74±0.09 | 2.08±0.10 | 2.17±0.10 | 2.39±0.09 | 2.51±0.09 |
| p | <0.001 | <0.001 | <0.001 | <0.001 | <0.001 | <0.001 | <0.001 |
| t | −8.04 | 8.32 | 26.66 | 4.50 | 18.08 | 17.60 | 7.72 |
| Atlas Name | Schaefer-600 | Schaefer-700 | Schaefer-800 | Schaefer-900 | Schaefer-1000 | Vosdewael-100 | Vosdewael-200 |
| Share Nbr | 2.48±0.14 | 2.76±0.14 | 2.85±0.16 | 2.86±0.12 | 3.07±0.14 | 1.57±0.17 | 1.71±0.11 |
| Unique Nbr | 2.60±0.10 | 2.74±0.10 | 2.74±0.12 | 2.87±0.09 | 3.03±0.10 | 1.46±0.10 | 1.73±0.08 |
| p | <0.001 | 0.39 | <0.001 | <0.001 | <0.001 | <0.001 | <0.001 |
| t | −14.04 | 2.42 | 11.98 | −5.75 | 4.23 | 34.09 | 7.44 |
| Atlas Name | Vosdewael-300 | Vosdewael-400 | Yeo2011(17) | Aparc | Aparc2009 | BA | Cole-Anticevic |
| Share Nbr | 1.96±0.12 | 2.21±0.15 | 1.76±0.11 | 1.58±0.12 | 1.95±0.13 | 1.58±0.12 | 1.65±0.11 |
| Unique Nbr | 2.02±0.09 | 2.32±0.10 | 1.73±0.08 | 1.33±0.07 | 1.94±0.09 | 1.29±0.08 | 1.57±0.07 |
| p | <0.001 | 0.13 | <0.001 | <0.001 | <0.001 | <0.001 | <0.001 |
| t | 5.41 | −2.82 | 22.29 | 56.37 | 3.80 | 69.84 | 22.44 |

# Appendix 8

## List of Genes Selected by Lasso

**Appendix 8—table 1.** The 28 genes selected by LASSO and their corresponding p-values from Welch's t-test.

| Gene Symbol | p | Gene Symbol | p | Gene Symbol | p | Gene Symbol | p |
|---|---|---|---|---|---|---|---|
| INPP4A | 0.76 | TLR1 | 0.02 | KCNH5 | 0.04 | OTULIN | 0.18 |
| ITGA1 | 0.19 | TPST1 | 0.94 | TMEM248 | 0.27 | DTX2 | 0.15 |
| JUNB | 0.57 | SNAP29 | 0.01 | ANO2 | 0.26 | SERPINB9P1 | 0.12 |
| PECAM1 | 0.04 | TRAM2 | 0.70 | PLEKHA3 | 0.90 | LHFPL5 | 0.63 |
| PRKCH | 0.10 | DHRS4 | 0.05 | PLBD1 | 0.01 | GK5 | 0.51 |
| NECTIN1 | 0.84 | LPIN1 | 0.34 | DENND1C | 0.37 | ZNF662 | 0.77 |
| SRC | 0.20 | BHMT2 | 0.01 | CXXC4 | 0.20 | NAP1L6 | 0.58 |

