## [Editor Report · eLife assessment]

This **important** paper compares cross-species cortical folding patterns in human and non-human primates, showing that most gyral peaks shared across species are in lower-order cortical regions. The supporting evidence is **solid** and multi-faceted, encompassing anatomy, connectivity and gene expression. This paper will be of interest to a broad readership within the neuroscience community, especially for those interested in cross-species correspondences in brain organisation.

---

## [Referee Report · Reviewer #1 (Public Review)]

Zhang et al. tackle the important topic of primate-specific structural features of the brain and the link with functional specialization. The authors explore and compare gyral peaks of the human and macaque cortex through non-invasive neuroimagery, using convincing techniques that have been previously validated elsewhere. They show that nearly 60% of the macaque peaks are shared with humans, and use a multi-modal parcellation scheme to describe the spatial distribution of shared and unique gyral peaks in both species.

The claim is made that shared peaks are mainly located in lower-order cortical areas whereas unique peaks are located in higher-order regions, however, no systematic comparison is made. The authors then show that shared peaks are more consistently found across individuals than unique peaks, and show a positive but small and non-significant correlation between cross-individual counts of the shared peaks of the human and the macaque i.e. the authors show a non-significant trend for shared peaks that are more consistently found across humans to be those that are also more found across macaques.

In order to identify if unique and shared peaks could be identified based on the structural features of the cortical regions containing them, the authors compared them with t-tests. A correction for multiple comparisons should be applied and t-values reported. Graph-theoretical measures were applied to functional connectivity datasets (resting-state fMRI) and compared between unique and shared peak regions for each species separately. Again the absence of multiple comparison correction and t-values make the results hard to interpret. The same comment applies to the analysis reporting that shared peaks are surrounded by a larger number of brain regions than unique peaks. Finally, the potentially extremely interesting results about differential human gene expression of shared and unique peaks regions are not systematically reported e.g. the 28 genes identified are not listed and the selection procedure of 7 genes is not fully reported.

The paper is well written and the methods used for data processing are very compelling i.e. the peak cluster extraction pipeline and cross-species registration.

Comments on revision:

The authors have convincingly addressed all my previous concerns such that, as the revised paper stands now, the presented results provide solid support for the conclusions of the authors. The revised paper is now of interest for a large part of the neuroscience community and specifically for those interested in primate-specific structural features of the brain and the link with functional specialization.

---

## [Author Response]

The following is the authors’ response to the original reviews.

**Public review**

**Reviewer 1**
Zhang et al. tackle the important topic of primate-specific structural features of the brain and the link with functional specialization. The authors explore and compare gyral peaks of the human and macaque cortex through non-invasive neuroimagery, using convincing techniques that have been previously validated elsewhere. They show that nearly 60% of the macaque peaks are shared with humans, and use a multi-modal parcellation scheme to describe the spatial distribution of shared and unique gyral peaks in both species.

We thank the reviewer for his/her summary and affirmation of our work.

The claim is made that shared peaks are mainly located in lower-order cortical areas whereas unique peaks are located in higher-order regions, however, no systematic comparison is made. The authors then show that shared peaks are more consistently found across individuals than unique peaks, and show a positive but small and non-significant correlation between cross-individual counts of the shared peaks of the human and the macaque i.e. the authors show a non-significant trend for shared peaks that are more consistently found across humans to be those that are also more found across macaques.Answer: We appreciate the reviewer for raising questions about our work. In order to provide a more systematic comparison for the conclusion that ‘shared peaks are mainly located in lowerorder cortical areas whereas unique peaks are located in higher-order regions’, we have conducted two additional experiments. Following the reviewers’ suggestions, we conducted a statistical analysis of the ratio of shared and unique peaks within different brain networks (as depicted in Figure 2 (b)), and also presented the specific distribution quantities of the two types of peaks in both low- and high-order brain networks (as detailed in the corresponding Table 1). Through these three experiments, we have obtained a more systematic and comprehensive conclusion that ‘shared peaks are more distributed in lower-order networks, while unique peaks are more in higher-order networks’.In order to identify if unique and shared peaks could be identified based on the structural features of the cortical regions containing them, the authors compared them with t-tests. A correction for multiple comparisons should be applied and t-values reported. Graph-theoretical measures were applied to functional connectivity datasets (resting-state fMRI) and compared between unique and shared peak regions for each species separately. Again the absence of multiple comparison correction and t-values make the results hard to interpret. The same comment applies to the analysis reporting that shared peaks are surrounded by a larger number of brain regions than unique peaks. Finally, the potentially extremely interesting results about differential human gene expression of shared and unique peaks regions are not systematically reported e.g. the 28 genes identified are not listed and the selection procedure of 7 genes is not fully reported.

Answer: We appreciate the reviewer for their suggestions about the statistical analysis in our manuscript. Firstly, we applied False Discovery Rate (FDR) correction to all experiments involving multiple comparisons throughout the entire manuscript, and the corrected t-values are reported (Table 2-5 and A5-A6). Additionally, in response to the reviewers’ guidance regarding the gene analysis section, we provided a list of 28 genes (Table A7) selected by lasso, along with the t-values obtained from Welch’s t-test for the expression of the two type of peaks. The functions corresponding to the seven genes with final t-values below 0.05 are reported in Table 6.

The paper is well written and the methods used for data processing are very compelling i.e. the peak cluster extraction pipeline and cross-species registration. However, the analysis and especially the reporting of statistics, as they stand now, constitutes the main weakness of the paper. Some aspects of the statistical analysis need to be clarified.
**Reviewer 2**
The authors compared the cortical folding of human brains with folding in macaque monkey brains to reveal shared and unique locations of gyral peaks. The shared gyral peaks were located in cortical regions that are functionally similar and less changed in humans from those in macaques, while the locations of unique peaks in humans are in regions that have changed or expanded functions. These findings are important in that they suggest where human brains have changed more than macaque brains in their subsequent evolution from a common ancestor. The massive analysis of comparative results provides evidence of where humans and macaques are similar or different in cortical markers, as well as noting some of the variations within each of the two primates.

Answer: Gratitude to the reviewer for his/her summary and appreciation of our cross-species work.

Strengths:The study includes massive detail.Weaknesses:The manuscript is too long and there is not enough focus on the main points.

Answer: We appreciate the reviewer for pointing out the shortcomings in our manuscript. Firstly, considering the manuscript is too long, we have chosen to retain only the core experiments and relevant analyses in the main text. Relatively minor conclusions have been moved to the supplementary information, such as original Table 1 is now moved to the Supplementary Information as Table A1 (locations of all shared clusters). Additionally, some non-essential expressions in the original manuscript have been removed.

Our experiments primarily revealed the existence of partially shared cortical landmarks, known as gyral peaks, in both humans and macaques. We found that these shared and unique peaks are mainly distributed across low- and high-order brain networks. To emphasize this main point, we added two experiments on top of the existing ones to provide a more systematic explanation of this conclusion. We conducted a statistical analysis of the ratio of shared and unique peaks within different brain networks (as depicted in Figure 2 (b)), and also presented the specific distribution quantities of the two types of peaks in both low- and high-order brain networks (as detailed in the corresponding Table 1). By combining the results of these two experiments with the original manuscript’s statistical findings on the proportions of the two type of peaks in different brain networks, the conclusion that ‘shared and unique peaks are predominantly located in low-order and high-order brain networks’ becomes more prominent.

A brief listing of previous views on why fissures form and what factors are important would be helpful.

Answer: In response to this suggestion from the reviewer, we have incorporated some previous views on why fissures form and what factors are important into the ‘Introduction’ section.

‘Cortical folds are important features of primate brains. The primary driver of cortical folding is the differential growth between cortical and subcortical layers. During the gyrification process in the cortex, areas with high-density stiff axonal fiber bundles towards gyri. The brain’s folding pattern formed through a series of complex processes. The folding patterns in the brain, formed through a series of complex processes, are found to play a crucial role in various cognitive and behavioral processes, including perception, action, and cognition (Fornito et al. 2004; Cachia et al. 2018; Yang et al. 2019; Whittle et al. 2009).’

**Reviewer 1 (Recommendations For The Authors):**
(1) Figure 3b shows a non-significant trend for shared peaks that are more consistently found across humans to be those that are also more found across macaques. In the discussion, lines 218-219, the fact that the correlation is not significant should be reported more clearly.

Answers: We thank the reviewer for this question. We revised the Line 218-219 (now Line 257-259) as follows: ‘2. Consistency: The inter-individual consistency of shared peaks within each species was greater than that of unique peaks. The consistency of shared peaks in the human and macaque brains exhibits a positive correlation (non-significant though).’

(2) It is not fully clear how much shared peaks are mostly distributed in the higher-order cortex, especially in the macaque. It is reported in the results lines 132-133 that ‘In the macaque brain, shared peak cluster centers most distributed in the V2, DMN, and CON (Figure.2 (d)), while unique peak cluster centers most distributed in the DMN, Language (Lan), and Dorsal-attention (DAN)’ but not further discussed. Please develop this point in the discussion. Further, the results presented in Figures 2 and A1 are actually quite different and this shall be better described in the results. Given that shared and unique peaks can be found in the same region, this analysis would gain importance by applying a comparison test for the selection of regions where the most shared or unique peaks are found. The sentence lines 306-308 should be accordingly revised.It is hard to understand what the 0-3% corresponds to in Figures 2 and A1?Please also correct in both legends and in the text the labeling of panels and add in the legends a brief description of panel (c). In the legend of Figure 2, ‘shared peaks’ in the second sentence shall be replaced by ‘unique peaks’.

Answers: We thank the reviewer for these questions and suggestions. Our responses to them are itemized as follows:

A1: In general, to clarify the distribution of shared and unique peaks in the high-order and loworder networks, we divided 12 brain networks in Cole-Anticevic atlas into the low-order networks (visual 1 (V1), visual 2 (V2), auditory (Aud), somatomotor (SMN), posterior multimodal (PMN), ventral multimodal (VMN), and orbito-affective networks (OAN)) and higher-order networks (include cingulo-opercular (CON), dorsal attention (DAN), language (Lan), frontoparietal (FPN), default mode network (DMN)) based on previous research (Golesorkhi et al. 2022; Ito, Hearne, and Cole 2020). On this lower/higher -order division, we reported the number of shared and unique peaks in both species in Author response table 1. It is found that, whether in humans or macaques, shared peaks are more distributed in lower-order networks, while unique peaks are more in higher-order networks. This observation is particularly pronounced in humans.

**Author response table 1. sa2table1:** The number of shared and unique peaks in lower- and higher-order brain networks of the two species. Lower-order networks include visual 1 (V1), visual 2 (V2), auditory (Aud), somatomotor (SMN), posterior multimodal (PMN), ventral multimodal (VMN), and orbito-affective networks (OAN), higher-order networks include cingulo-opercular (CON), dorsal attention (DAN), language (Lan), frontoparietal (FPN), default-mode network (DMN).

Lower/Highernetworks	Human	Macaque
Shared peak	33//18	29//22
Unique peak	37//104	14//20

In the main text, Figure 2 (referring to Author response figure 1 later in the text.) illustrates the proportions of shared and unique peaks across 12 brain networks in both species. In each pie chart, we have specifically highlighted the top three ranked brain regions. Although the pie chart also generally supports the above results, two brain networks deserve further discussion. They are DMN and CON, two higher-order networks that have higher ranks in terms of shared peak count (the second-ranked and the third-ranked on macaque shared peaks; the fourth-ranked and the fifth-ranked on human shared peaks).

The cingulo-opercular network (CON) is a brain network associated with action, goal, arousal, and pain. However, a study found three newly discovered areas of the primary motor cortex that exhibit strong functional connectivity with the CON region, forming a novel network known as the somato-cognitive action network (SCAN) (Gordon et al. 2023). The SCAN integrates body control (motor and autonomic) and action planning, consistent with the findings that aspects of higher-level executive control might derive from movement coordination (Llinás 2002; Gordon et al. 2023). CON may be shared in the form of the SCAN network across these two species. This could explain in part the results in Author response figure 1 that shared peaks are more on CONs.

**Author response image 1. sa2fig1:** Pie chart shows the count of shared and unique peaks across different brain networks for both human and macaque. Right panel shows the Cole-Anticevic (CA) networks (Ji et al. 2019) on human surface as a reference.

Default-mode network (DMN) is a ensemble of brain regions that are active in passive tasks, including the anterior and posterior cingulate cortex, medial and lateral parietal cortex, and medial prefrontal cortex (Buckner, Andrews-Hanna, and Schacter 2008). Although DMN is considered a higher-order brain network, numerous studies have provided evidence of its homologous presence in both humans and macaques. Many existing studies have confirmed the similarity between the DMN regions in humans and macaques from various perspectives, including cytoarchitectonic (Parvizi et al. 2006; Buckner, Andrews-Hanna, and Schacter 2008; Caminiti et al. 2010) and anatomical tracing (Vincent et al. 2007). These studies all support the notion that some elements of the DMN may be conserved across primate species (Mantini et al. 2011). In general, the partial sharing of DMN between humans and macaques may be attributed to the higher occurrence of shared peaks within the DMN.

These results have been added to Table 2 along with corresponding text and discussion section.

A2: The difference between the results of Figure 2 and Figure A1 (now Figure A2) is whether the peak count is normalized by cortical area, which hugely varies across networks. For example, among the 12 brain networks, the three networks with the largest surface areas are the DMN, SMN and CON, and the three networks with the smallest area are OAN, PMN and VMN. The area difference between networks can be as large as 18-fold. Therefore, it is not difficult to find that, although the DMN ranks high in both shared and unique peak counts during statistical analysis (Figure 2 (a)), it is relatively small in Figure A2 after area normalization. In contrast, VMN ranks lower in peak count statistics but exhibits a substantial proportion after area normalization (For example, 38% of macaque shared peaks are distributed in the VMN region, but there are actually only four peaks). However, the two pie charts deliver the same message that there are more shared peaks in lower-order networks, while unique peaks are more in higher-order networks (except for macaques, where shared peaks are also distributed significantly in DMN and CON).

Following the suggestion from the reviewer, we adopted a new approach to present the ratio between shared peak count and unique peak count for each network (see Author response figure 2), such that the networks where the most shared or unique peaks are found can be easily highlighted. To mitigate potential imbalances in proportions caused by differences in the absolute numbers of each category (shared or unique) of peak, the proportions of peaks within their respective categories were utilized in the calculations. In Author response figure 2, the pink and green color bins represent ratios of shared and unique peaks, respectively. The dark blue dashed line represents the 50% reference line. In general, from left to right in the figure, the ratio of shared peaks decreases gradually while the ratio of unique peaks increases, suggesting that shared peaks are more (>0.5, above the dashed line) on lower-order networks (orange font), while unique peaks are generally more on higher-order networks (blue font). In specific, in human brains, the networks with a higher abundance of shared peaks are Aud, VMN, V1, SMN, and V2; whereas in macaques, they are CON, VMN, V1, V2, FPN, and SMN. Again, in the human brains, the disparity between shared and unique peaks tends to be more significant (further away from the reference line), for both lower-order and higher-order networks, respectively. In contrast, in the macaque brains, the disparity between shared and unique peaks is less significant (closer to the reference line). The ratio of shared and unique peaks is around 0.5 for 6 out of all 10 networks (including both lower and higher-order ones).

**Author response image 2. sa2fig2:** The ratio of shared and unique peaks in each brain network in the Cole-Anticevic (CA) atlas. The pink and green color bins represent ratios of shared and unique peaks, respectively. The dark blue dashed line represents the 50% reference line. For each brain region, the sum of the ratios of shared and unique peaks is equal to 1.

Based on these analyses, the sentence lines 306-308 (now Line 368-370) has been revised as follows: ‘In the human brain, the more shared peaks (about 65%) are located in lower-order brain regions, while unique peaks are mainly (about 74%) located in higher-order regions. However, this trend is relatively less pronounced in the macaque brain.’

These results have been added to Figure 2 (b) along with corresponding text and discussion section.

A3: In response to the third suggestion from the reviewer, we have clearly labeled the brain region names corresponding to 0% to 3% in Figure 2 (now Figure 2 (a)) and Figure A1 (now Figure A2).

**Author response image 3. sa2fig3:** Pie chart shows the count of shared and unique peaks across different brain networks for both human and macaque. Right panel shows the Cole-Anticevic (CA) networks (Ji et al. 2019) on human surface as a reference.

A4: Finally, we would like to express our gratitude to the reviewer for pointing out our mistakes.

We have made improvements to Figure 2 and revised the figure captions accordingly.

(3) The conclusions regarding the spatial relationship between peaks and functional regions shall be revised (Lines 187-188, 228-229, and 329-330). In the macaque, the results are opposite in the two atlases used. Further, in the human, it is not clear how multiple comparison corrections will impact statistics and some atlases show opposite results, although conclusions hold true in the majority of human atlases.

Answers: We thank the reviewer very much for this suggestion. We have added the results of the Cole-Anticevic atlas for macaques in the main text, which also has the observation that shared>unique (Author response table 2, corresponds to Table 5 in main text), namely, there are more diverse brain regions around shared peaks than around unique peaks. Therefore, out of the commonly used three macaque atlases, two (Markov91 and Cole-Anticevic) conform to this observation, while BA05 does not. We utilized false discovery rate (FDR) correction for multiple comparisons, and the corrected p-values are reported in Tables (in the revised main text and are shown below). Results on atlas with multiple resolutions are reported in Author response table 4 (Table A6 in the Supplementary Information). The observation that more diverse brain regions around shared peaks than around unique peaks, holds for human atlases in Author response table 3 (Table 4 in main text), where the atlas resolutions ranges from 7 parcels to 300 parcels, demonstrating the robustness of the conclusion. It is noted that the observation is not consistent on atlases with relatively lower resolutions (e.g., BA05 for macaque, n=30 and Yeo2011 for human, n=7) or, in particular, higher resolutions (e.g., Schaefer-500, and Vosdewael-400, n>300). This inconsistency could be reasonable since the resolution of the parcellation itself will largely determines the chance of a cortical region appear in a peak’s neighborhood, if the parcellation is too coarse or too fine. For example, if n=1 (the entire cortex is the only one region) or n=30k (each vertex is a region), each peak will has the same number of neighboring regions for these two extreme cases (one brain region for each peak for n=1; around 30 vertices for each peak for n=30k).

In conclusion, we observed that there are more diverse brain regions around shared peaks than around unique peaks for multiple brain atlases with a median parcellation resolution. These results have been added to Tables 4, 5, and A6 along with corresponding text and discussion section.

**Author response table 2. sa2table2:** The mean values (± SD) of brain regions that appeared within a 3-ring neighborhood for shared and unique peaks in 3 common macaque atlases. For both Markov91 and Cole-Anticevic atlas, the shared peaks has more variety of functional regions around it than the unique peaks. But for the altas BA05, the conclusion was reversed. The bold font represent the larger values between the shared peak and unique peaks. All p<0.001, after false discovery rate (FDR) corrected.

AtlasName	Markov91	Cole-Anticevic	BA05
Share	2.73+-0.27	1.77+-0.17	1.61+-0.16
Nbr			
Unique	2.16+-0.15	1.58+-0.16	1.80+-0.16
Nbr			
p	< 0.001	< 0.001	< 0.001
t	-7.4	14.93	6.49

(4) For Tables 2-4, A4, and Figure 3a, please indicate in all the legends if values correspond to Mean plus minus Standard Deviation, report t-value, and n in the legend or in the text.

Answers: We thank the reviewer very much for this suggestion. We added the ‘mean (± SD)’ in the notes of Tables 2-4, A4 (now A6), and Figure 3 (a). All the t and n values of t-test are reported in tables or in the main text.

(5) Please create a statistical section in the Methods to describe more precisely the tests used e.g. for t-tests, if datasets follow a normal distribution with unknown variance. In the case of multiple comparisons like in e.g. Table 2-4, A4, please report what multiple comparisons correction was used to adjust the significance level.

**Author response table 3. sa2table3:** The mean values (± SD) of brain regions that appeared within a 3-ring neighborhood for shared and unique peaks in 10 common human atlases. All the shared peaks in the table have a greater number of neighboring brain regions compared to the unique peaks. All p<0.001, false discovery rate (FDR) corrected.

AtlasName	Glasser2016	Schaefer-100	Schaefer-200	Schaefer-300	Vosdewael-100
ShareNbr	2.43+-0.15	1.89+-0.12	2.12+-0.11	2.23+-0.11	1.57+-0.17
UniqueNbr	2.37+-0.09	1.74+-0.09	2.08+-0.10	2.17+-0.10	1.46+-0.10
p	< 0.001	< 0.001	< 0.001	< 0.001	< 0.001
t	8.32	26.66	4.50	18.08	34.09
AtlasName	Yeo2011(17)	Aparc	Aparc2009	BA	Cole-Anticevic
ShareNbr	1.76+-0.11	1.58+-0.12	1.95+-0.13	1.58+-0.12	1.65+-0.11
UniqueNbr	1.73+-0.08	1.33+-0.07	1.94+-0.09	1.29+-0.08	1.57+-0.07
p	< 0.001	< 0.001	< 0.001	< 0.001	< 0.001
t	22.29	56.37	3.80	69.84	22.44

**Author response table 4. sa2table4:** The mean values (± SD) of brain regions where shared and unique peaks appeared within a 3-ring neighborhood in 21 common human atlases. The p-values were corrected by FDR.

AtlasName	Yeo2011(7)	Glasser2016	Schaefer-100	Schaefer-200	Schaefer-300	Schaefer-400	Schaefer-500
Share Nbr	1.48+-0.10	2.43+-0.15	1.89+-0.12	2.12+-011	2.23+-011	2.46+-013	2.50+-0.14
UniqueNbr	1.54+-0.07	2.37+-0.09	1.74+-0.09	2.08+-010	2.17+-0.10	2.39+-0.09	2.51+-009
p	< 0.001	< 0.001	< 0.001	< 0.001	< 0.001	< 0.001	< 0.001
	-8.04	8.32	26.66	4.50	18.08	17.60	7.72
AtlasName	Schaefer-600	Schaefer-700	Schaefer-800	Schaefer-900	Schaefer-1000	Vosdewael-100	Vosdewael-200
Share Nbr	2.48+-0.14	2.7	2.85+-0.16	2.86+-012	3.07+-014	1.57+-017	1.71+-0.11
UniqueNbr	2.60+-0.10	2.74+-0.10	2.74+-0.12	2.87+-009	3.03+-0.10	1.46+-0.10	1.73+-008
p	< 0.001	0.39	< 0.001	< 0.001	< 0.001	< 0.001	< 0.001
	-14.04	2.42	11.98	-5.75	4.23	34.09	7.44
AtlasName	Vosdewael-300	Vosdewael-400	Yeo2011(17)	Aparc	Aparc2009	BA	Cole-Anticevic
Share Nbr	1.96+-0.12	2.21+-0.15	1.76+-0.11	1.58+-012	1.95+-013	1.58+-0.12	1.65+-011
UniqueNbr	2.02+-0.09	2.32+-0.10	1.73+-0.08	1.33+-007	1.94+-0.09	1.29+-0.08	1.57+-0.07
n	< 0.0015.41	0.13-2.82	< 0.00122.29	< 0.00156.37	< 0.0013.80	< 0.00169.84	< 0.00122.44

Answers: Thanks for the reviewer’s suggestion, we added a ‘Statistic Analysis’ section in the ‘Materials and Methods’ part:

‘All variables used in the two-samples t-test follow a normal distribution check and all p-values were corrected for multiple comparisons using the false discovery rate (FDR) method. Moreover, in order to identify differently expressed genes between shared and unique peaks, we employed the Welch’s t-test, given the unequal sample sizes for shared and unique peaks. For all tests, a p-value <0.05 was considered significant (FDR corrected).’

For the experiments of multiple comparisons such as Table 2-4, A4 (now A6), etc., we have added explanations in the main text, multiple comparisons correction has been corrected by false discovery rate (FDR), p-value<0.05 is considered significant.

(6) It would be of great interest to provide the full list of the 28 genes that significantly contributed to the classification of shared and unique peaks. Please provide a description of the Welch’s t-test results. From the 7 genes selected, only two are discussed. Could the authors please describe briefly the function of the other genes? Although we understand that they are not associated with neuronal activity and brain function.

Answers: We thank the reviewer for these suggestions. We have provided a complete list of 28 genes selected by LASSO in the Author response table 5. Additionally, Welch’s t-test was employed to calculate p-values for the expression differences of each gene in shared and unique peak clusters, and the results are also reported in the Author response table 5.

**Author response table 5. sa2table5:** The 28 genes selected by LASSO and their corresponding p-values from Welch’s t-test.

GeneSymbol	p	GeneSymbol	p	GeneSymbol	p	GeneSymbol	p
INPP4A	0.76	TLR1	0.02	KCNH5	0.04	OTULIN	0.18
ITGA1	0.19	TPST1	0.94	TMEM248	0.27	DTX2	0.15
JUNB	0.57	SNAP29	0.01	ANO2	0.26	SERPINB9P1	0.12
PECAM1	0.04	TRAM2	0.70	PLEKHA3	0.90	LHFPL5	0.63
PRKCH	0.10	DHRS4	0.05	PLBD1	0.01	GK5	0.51
NECTIN1	0.84	LPIN1	0.34	DENND1C	0.37	ZNF662	0.77
SRC	0.20	BHMT2	0.01	CXXC4	0.20	NAP1L6	0.58

Seven genes showed significant differential expression between shared and unique peaks in Welch’s t-test. These genes were PECAM1, TLR1, SNAP29, DHRS4, BHMT2, PLBD1, KCNH5. Brief descriptions of their functions are listed in Author response table 6. All gene function descriptions were derived from the NCBI website (https://www.ncbi.nlm.nih.gov/).

These results have been added to Tables 6 and A7 along with corresponding text.

(6) For comparison, could the authors provide a supplementary figure of shared peak clusters like in Figure 1b but displayed on the surface of the macaque brain template?

Answers: We thank the reviewer very much for this suggestion and we have incorporated a display of shared peak clusters on the macaque brain template surface (Author response figure 4, corresponds to Figure A1 of Supplementary Information.)

(7) Could the author develop or rephrase the sentence lines 69-72 which remains unclear?

Answers: We appreciate the reviewer’s feedback and have revised this sentence to ensure clarity. The sentences from line 69 to 72 have been revised to ‘In the study of macaques, it has been observed that the peak consistently present across individuals is located on more curved gyri (S. Zhang, Chavoshnejad, et al. 2022). Similar conclusions have been drawn in human brain research (S. Zhang, T. Zhang, et al. 2023).’ Now, this sentence corresponds to lines 74-77 in the main text.

(8) Line 99: please indicate which section.

**Author response table 6. sa2table6:** Seven genes were selected using LASSO that showed significant differential expression in shared and unique peaks.

GeneSymbol	ene Function
ECAM1	The protein encoded by this gene is found on the surface of platelets,monocytes, neutrophils, and some types of T-cells, and makes up a largeportion of endothelial cell intercellular junctions. The encoded protein is amember of the immunoglobulin superfamily and is likely involved inleukocyte migration, angiogenesis, and integrin activation. [provided byRefSeq, May 2010]
LR1	The protein encoded by this gene is a member of the Toll-like receptor (TLR)family which plays a fundamental role in pathogen recognition and activationof innate immunity. They recognize pathogen-associated molecular patterns(PAMPs) that are expressed on infectious agents, and mediate the productionof cytokines necessary for the development of effective immunity. [providedby RefSeq, Jul 2008]
NAP29	This gene, belonging to the SNAP25 gene family, encodes a protein involvedin various membrane trafficking processes. Other members of this genefamily, such as SNAP23 and SNAP25, encode proteins that bind to a syntaxinprotein and facilitate the docking and fusion of synaptic vesicle membraneswith the plasma membrane. [provided by RefSeq, Jul 2008]
HRS	Exhibits protein binding and oxidoreductase activities, involved in cellularmetabolic processes including ketone metabolism, regulation of reactiveoxygen species, and steroid metabolism. Found in the nucleus andperoxisomal membrane. [provided by Alliance of Genome Resources, Apr2022 ]
HMT2	Homocysteine, a sulfur-containing amino acid, is crucial for methylationreactions. The protein encoded by this gene is one of two methyltransferasesthat facilitate the transfer of a methyl group from betaine to homocysteine.Irregularities in homocysteine metabolism have been linked to conditionsranging from vascular disease to neural tube birth defects. This gene hasalternatively spliced transcript variants encoding differentisoforms.[provided by RefSeq, May 2010]
LBD1	Predicted to enable phospholipase activity. Predicted to be involved inphospholipid catabolic process. Located in extracellular space. [provided byAlliance of Genome Resources, Apr 2022]
CNH5	This gene encodes a member of voltage-gated potassium channels. Membersof this family have diverse functions, including regulating neurotransmitterand hormone release, cardiac function, and cell volume. This protein is anoutward-rectifying, noninactivating channel. Alternative splicing results inmultiple transcript variants. [provided by RefSeq, Jul 2013]

Answers: We thank the reviewer very much for this suggestion and we revised this sentence to ‘The definition of peaks and the method for extracting peak clusters within each species are described in the Materials and Methods section’.

(9) In Figure 3b, please report R2 and p-value. A semi-log might be more appropriate given the overdispersion of Human Peak Counts.

Answers: We thank the reviewer very much for this suggestion. Linear regression analysis was conducted on the average counts of all corresponding shared peak clusters of human and macaque. The horizontal and vertical axes of the Author response figure 5 (b) represent the average count of shared peaks in the macaque and human brains, respectively. The Pearson correlation coefficient (PCC) of the interspecies consistency of the left and right brain is 0.20 and 0.26 (p>0.05 for both), respectively. The result of linear regression shows that there is a positive correlation in the inter-individual consistency of shared peaks between macaque and human brains, but it is not statistically significant (with R2 for the left and right brain are 0.07 and 0.01, respectively).

**Author response image 4. sa2fig4:** Shared peak clusters of macaque, shows on macaque brain template.

The goodness of fit (R2), pearson correlation coefficient (PCC), and their respective p-values were indicated in Author response figure 5 (b). To avoid overdispersion, the peak count of the human brain is displayed in a semi-log format.

The updated Figure and results are presented in Figure 3 of the main text.

(10) Line 177: please indicate where in the Supplementary Information.

Answers: Thank you for the reviewer’s reminder. We have incorporated the results of the human brain structural connectivity matrix into Table A5 in the Supplementary Information and provided corresponding indications in the main text.

(11) Line 226: please correct ‘except for betweeness [and efficiency] of the’.

Answers: We thank the reviewer very much for this suggestion and we added ‘and efficiency’ in original Line 173 and 226 (now Line 206 and 267) after ‘betweeness’.

(12) The gene expression dataset used is from the Allen Human Brain Atlas (AHBA). Reference to Hawrylycz et al., 2012 Nature. 2012 Sep 20;489(7416):391-399. doi: 10.1038/nature11405 shall be made and abbreviation defined at first use in the text.

Answers: We added the full name ‘Allen Human Brain Atlas’ when AHBA is first mentioned, along with the reference suggested by the reviewer.

**Author response image 5. sa2fig5:** Consistency results of shared and unique peaks of two species. (a) Mean peak count (± SD) covered by shared and unique peak clusters in two species.***indicates p<0.001. The t-values for the t-tests in humans and macaques are 4.74 and 2.67, respectively. (b) Linear regression results of the consistency of peak clusters shared between macaque and human brains. The pink and blue colors represent the left and right hemispheres, respectively. The results of the linear regression are depicted in the figure. While there was a positive correlation observed in the consistency of gyral peaks between macaque and human, the obtained p-value for the fitted results exceeded the significance threshold of 0.05.

(13) Line 17: remove ‘are’.

Answers: We thank the reviewer very much for this suggestion and we removed ‘are’ in Line 17 (now Line 18).

(14) Line 201: remove ‘is used’.

Answers: We thank the reviewer very much for this suggestion and we removed ‘is used’ in Line 201 (now Line 237).

References

Buckner, Randy L, Jessica R Andrews-Hanna, and Daniel L Schacter (2008). “The brain’s default network: anatomy, function, and relevance to disease”. In: Annals of the new York Academy of Sciences 1124.1, pp. 1–38.

Cachia, Arnaud et al. (2018). “How interindividual differences in brain anatomy shape reading accuracy”. In: Brain Structure and Function 223, pp. 701–712.

Caminiti, Roberto et al. (2010). “Understanding the parietal lobe syndrome from a neurophysiological and evolutionary perspective”. In: European Journal of Neuroscience 31.12, pp. 2320–2340.

Fornito, Alexander et al. (2004). “Individual differences in anterior cingulate/paracingulate morphology are related to executive functions in healthy males”. In: Cerebral cortex 14.4, pp. 424–431.

Golesorkhi, Mehrshad et al. (2022). “From temporal to spatial topography: hierarchy of neural dynamics in higher-and lower-order networks shapes their complexity”. In: Cerebral Cortex 32.24, pp. 5637–5653.

Gordon, Evan M et al. (2023). “A somato-cognitive action network alternates with effector regions in motor cortex”. In: Nature, pp. 1–9.

Ito, Takuya, Luke J Hearne, and Michael W Cole (2020). “A cortical hierarchy of localized and distributed processes revealed via dissociation of task activations, connectivity changes, and intrinsic timescales”. In: NeuroImage 221, p. 117141.

Ji, Jie Lisa et al. (2019). “Mapping the human brain’s cortical-subcortical functional network organization”. In: Neuroimage 185, pp. 35–57.

Llinás, Rodolfo R (2002). I of the vortex: From neurons to self. MIT press.

Mantini, Dante et al. (2011). “Default mode f brain function in monkeys”. In: Journal of Neuroscience 31.36, pp. 12954–12962.

Parvizi, Josef et al. (2006). “Neural connections of the posteromedial cortex in the macaque”. In:Proceedings of the National Academy of Sciences 103.5, pp. 1563–1568.

Vincent, Justin L et al. (2007). “Intrinsic functional architecture in the anaesthetized monkey brain”.In: Nature 447.7140, pp. 83–86.

Whittle, Sarah et al. (2009). “Variations in cortical folding patterns are related to individual differences in temperament”. In: Psychiatry Research: Neuroimaging 172.1, pp. 68–74.

Yang, Shimin et al. (2019). “Temporal variability of cortical gyral-sulcal resting state functional activity correlates with fluid intelligence”. In: Frontiers in neural circuits 13, p. 36.

Zhang, Songyao, Poorya Chavoshnejad, et al. (2022). “Gyral peaks: Novel gyral landmarks in developing macaque brains”. In: Human Brain Mapping 43.15, pp. 4540–4555.

Zhang, Songyao, Tuo Zhang, et al. (2023). “Gyral peaks and patterns in human brains”. In: Cerebral Cortex.